# Unveiling causal relationship between white matter tracts and psychiatric disorders
Yifan Yu [1], Tianye Jia [2], Xiao Lin[1], Yanping Bao [3], Suhua Chang[1], Jie Sun[4], Teng Gao[1], Jie Shi [3,5], Sizhi Ai [6,7,8] ✉ & Kai Yuan [1,8] ✉

White matter tracts (WMTs), which mediate information transmission in the brain, are closely associated with the pathogenesis of psychiatric disorders, yet the causality of their associations remain unclear. Thus, we employed two-sample bidirectional Mendelian randomization to explore the causality between WMTs and 10 psychiatric disorders. We found that one standard deviation changes of WMTs metrics modified risks for 8 psychiatric disorders by 2·2% to 71·4%. For example, increased fornix/stria terminalis radial diffusivities elevated PTSD risk by 8.3%, while heightened mode anisotropy reduced Tourette syndrome risk by 71.4%. Reversely, alcohol use disorder increased the risk of WMTs abnormalities. Our study provides novel insights into the potential causality between WMTs and psychiatric disorders, indicating that alterations of WMTs may serve as biomarkers for psychiatric disorders.

Psychiatric disorders impose a substantial burden on global health, as evidenced by the Global Burden of Diseases 2021[1], which shows that eight psychiatric disorders are among the top 25 contributors to global disease burden. Recently, a large body of literature has shown that white matter tracts (WMTs), which are responsible for information transmission, are closely related to the onset, development, and prognosis of psychiatric disorders[2–5]. Researchers often utilize diffusion tensor imaging (DTI) sequence of magnetic resonance imaging (MRI) to visualize WMTs and assess their integrity[6,7], as well as measure potential damage to axons[8], myelin[8], and cell membranes[8]. These functional changes in DTI[9,10] are characterized by five parameters/metrics: fractional anisotropy (FA), mean diffusivities (MD), axial diffusivities (AD), radial diffusivities (RD), and mode of anisotropy (MO). Different WMTs exhibit variations in both structure and function. For readers seeking further explanations, please refer to Table 1.

Research has indicated that certain characteristics of WMTs could potentially serve as diagnostic markers for psychiatric disorders[11]. For example, patients with major depressive disorder (MDD) who had suffered non-suicidal self-injury showed reduced WMT integrity compared to healthy controls[12] and appeared to have more severe dysfunction of the kynurenine/tryptophan pathway[13]. Compared with MDD, the alterations of WMTs in patients with bipolar disorder (BD) were more evident[14]. However, many studies focus solely on the FA value to represent changes in the properties of WMTs, as summarized in Table S1, while neglecting the contributions of other parameters such as MD, RD, AD, and MO. In addition, although many studies have shown associations between WMTs and psychiatric disorders, the causality and directions of these associations are still unclear. Therefore, the lack of clarity about the causal association between WMTs and psychiatric disorders, as well as the neglection of other important parameters, has limited the application of DTI techniques in the field of psychiatric disorders[9].

While causal inference in observational studies was commonly confounded by complex factors, such as the medications and environment[15], Mendelian randomization (MR) studies can effectively mitigate these confounding effects by utilizing genetic polymorphisms as instrumental variables (IVs) for exposure[16]. The core idea of MR is to "infer causality between exposure and outcome using genetic variants as IVs, leveraging the free and random assortment of genes and the stability of genotypes against

[1]Peking University Sixth Hospital, Peking University Institute of Mental Health, NHC Key Laboratory of Mental Health, National Clinical Research Center for Mental Disorders (Peking University Sixth Hospital), Chinese Academy of Medical Sciences Research Unit (No.2018RU006), Peking University, Beijing, China. [2]Institute of Science and Technology for Brain-Inspired Intelligence, Fudan University, Shanghai, China. [3]National Institute on Drug Dependence and Beijing Key Laboratory of Drug Dependence, Peking University, Beijing, China. [4]Center for Pain Medicine, Peking University Third Hospital, Beijing, China. [5]Peking University Health Science Center, Peking University, Beijing, China. [6]Center for Sleep and Circadian Medicine, The Affiliated Brain Hospital, Guangzhou Medical University, Guangzhou, China. [7]Key Laboratory of Neurogenetics and Channelopathies of Guangdong Province and the Ministry of Education of China, Guangzhou Medical University, Guangzhou, China. [8]These authors jointly supervised this work: Sizhi Ai, Kai Yuan. ✉e-mail: 2022760748@gzhmu.edu.cn; yuankai@pku.edu.cn

## Table 1 | The glossary of white matter tracts

| White matter tract | Type | Description |
|---|---|---|
| CST | Brainstem tract | This structure can be identified at medulla and the pons level, but should also contain corticopontine and corticobular tracts. This tract is a part of the left PLIC[33]. |
| CR | Projection tract | This structure is divided into three regions: ACR, SCR and PCR. The divisions are made at the middle of the genu and splenium of the corpus callosum, which are arbitrarily chosen and not based on anatomic or functional boundaries. This region includes the thalamic radiations (thalamocortical, corticothalamic fibers) and parts of the long corticofugal pathways, such as the corticospinal, corticopontine, and corticobulbar tracts. The boundary of the CR and the IC is defied at the axial level where the IC and EC merge. |
| ALIC | Projection tract | The anterior thalamic radiation and fronto-pontine fibers are the major contributors in this region. |
| PLIC | Projection tract | The superior thalamic radiation and long corticofugal pathways, such as the corticospinal tract and the fronto- and parieto-pontine fibers, are the major constitutes. |
| RLIC | Projection tract | In this region, the posterior thalamic radiation (corticothalamic and thalamo-cortical fibers, including the optic radiation) is the major constituent, but can also include the parieto-, occipito- and temporopontine fibers. The boundary with SS is arbitrarily defined at the middle of the SCC. |
| SLF | Association fiber | This tract is located at the dorsolateral regions of the CR and contains connections between the frontal, parietal, occipital, and the temporal lobes including language-related areas (Broca's, Geschwind'sand Wernicke's territories.) This tract is associated with working memory performance, attention[52] and language-related[53] functions. |
| SFO | Association fiber | This tract is located at the superior edge of the ALIC (anterior thalamic radiation) and the boundary is not always clear. Only the frontal region is identifiable and projection to the parietal lobe cannot be segmented. It has been suggested that this tract is a part of the anterior thalamic radiation and not an association fiber. |
| UNC | Association fiber | This tract connects the frontal lobe (orbital cortex) and the anterior temporal lobe. It can be discretely identified where the two lobes are connected but not within the frontal and the temporal lobes where it merges with other tracts. This tract is associated with language-related[53] function and its damage will lead to inattentive-emotional symptoms and cognitive deficits[54]. |
| IFO/UNC IFO/ILF | Association fiber | The IFO connects the frontal lobe and the occipital lobe. In the frontal lobe, this partition also includes the frontal projection of the UNC. In the temporal and occipital lobe, the IFO merges with the ILF, which is segmented as a different partition. The IFO is the longest association fiber running medially in the temporal lobe and connects the frontal lobe with the occipital, temporal and superior parietal cortex[33]. And it poses language-related[53] functions and its damage will cause inattention and emotional problems[54]. |
| SS | Association fiber | The IFO/ILF merges with projection fibers from the RLIC and forms a large, sheet-like, sagittal structure, called the SS. This region, therefore, should include both association and projection fibers. The boundary of the SS and the PCR is also arbitrarily defined at the axial level of the SCC. |
| EC | Association fiber | This region, located lateral to the IC, is believed to contain association fibers, such as the ALF and IFO and commissural fibers. Because of the limited image resolution, the external and extreme capsules are not resolved. |
| CG | Association fiber | This tract connects the frontal lobe with the amygdala[33], carrying information from the cingulate gyrus to the hippocampus. The entire pathway from the frontal lobe can be clearly identified. In the WMPM, the CG in the cingulate gyrus and the hippocampal regions is separated at the axial level of the SCC and denoted as CGC and CGH, respectively. It is associated with emotion and fear extinction[33]. |
| FX/ST | Association fiber | These tracts are both related to the limbic system: the FX to the hippocampus, and the ST to the amygdala. Both tracts project to the septum and the hypothalamus. With current image resolution capabilities, these two tracts cannot be distinguished in the hippocampal area, and both tracts are labeled as FX. The ST can be discretely identified in the amygdala and the dorsal thalamus. |
| BCC | Commissural fiber | The body of CC. This tract interconnects parietal and temporal cortices[52], connecting bilateral premotor, primary motor, and primary sensory cortex[55]. Its damage will cause sensory and motor processing abnormalities. |
| GCC | Commissural fiber | The genus of CC. This tract connects the bilateral frontal cortex and is involved in sensory and visuospatial processing[34,35]. Its damage will cause social functioning impairment[34] or disassociative symptoms[33]. |
| SCC | Commissural fiber | The splenium of CC. This tract receives input from the occipital lobes[52]. |

The definition and description of white matter tracts were from Mori et al.[10].

*ACR* anterior corona radiata, *ALIC* anterior limb of internal capsule, *ATR* anterior thalamic radiation, *BCC* body of corpus callosum, *CC* corpus callosum, *CGC* cingulum connecting to cingulate gyrus, *CGH* cingulum connecting to hippocampus, *CR* corona radiata, *CST* corticospinal tract, *EC* external capsule, *FX/ST* fornix and stria terminalis, *FX* Fornix (column and body of fornix), *GCC* genu of corpus callosum, *IC* internal capsule, *ICP* inferior cerebellar peduncle, *IFO/ILF* inferior fronto-occipital fasciculus/inferior longitudinal fasciculus, *IFO/UNC* inferior fronto-occipital fasciculus/uncinate fasciculus, *ILF* inferior longitudinal fasciculus, *PCR* posterior corona radiata, *PLIC* posterior limb of internal capsule, *PTR* posterior thalamic radiation (include optic radiation), *RLIC* retrolenticular part of internal capsule, *SCC* splenium of corpus callosum, *SCR* superior corona radiata, *SFO* superior fronto-occipital fasciculus, *SLF* superior longitudinal fasciculus, *SS* Sagittal stratum, *STR* superior thalamic radiation, *UNC* uncinate fasciculus.

environmental influences to avoid confounding bias in traditional epidemiology effectively"[17]. Current MR research has explored the causal association between imaging-derived phenotype and 10 psychiatric disorders[18], between brain functional networks and 12 psychiatric disorders[19], between white matter hyperintensities and anxiety disorders[20], between brain structure and Alzheimer's disease[21], and between cortical structure, white matter microstructure, and neurodegenerative diseases[22]. These studies revealed some causal associations between brain structure and function and neuropsychiatric disorders. However, these studies fail to uncover the causalities between WMTs and psychiatric disorders. Thus, we conducted bidirectional MR in this study to explore the causal associations between WMTs and psychiatric disorders. We found that the changes of WMTs

metrics modified risks for 8 psychiatric disorders. Reversely, alcohol use disorder (AUD) increased the risk of WMTs abnormalities. Our study provides novel insights into the potential causality between WMTs and psychiatric disorders, indicating that alterations of WMTs may serve as biomarkers for psychiatric disorders.

## Results
### Overview of MR
The research design is illustrated in Fig. 1. and the baseline characteristics of genome-wide association studies (GWAS) are described in Table 2. Among all included GWAS data of psychiatric disorders, only AUD and post-traumatic stress disorder (PTSD) included a small portion of UK Biobank

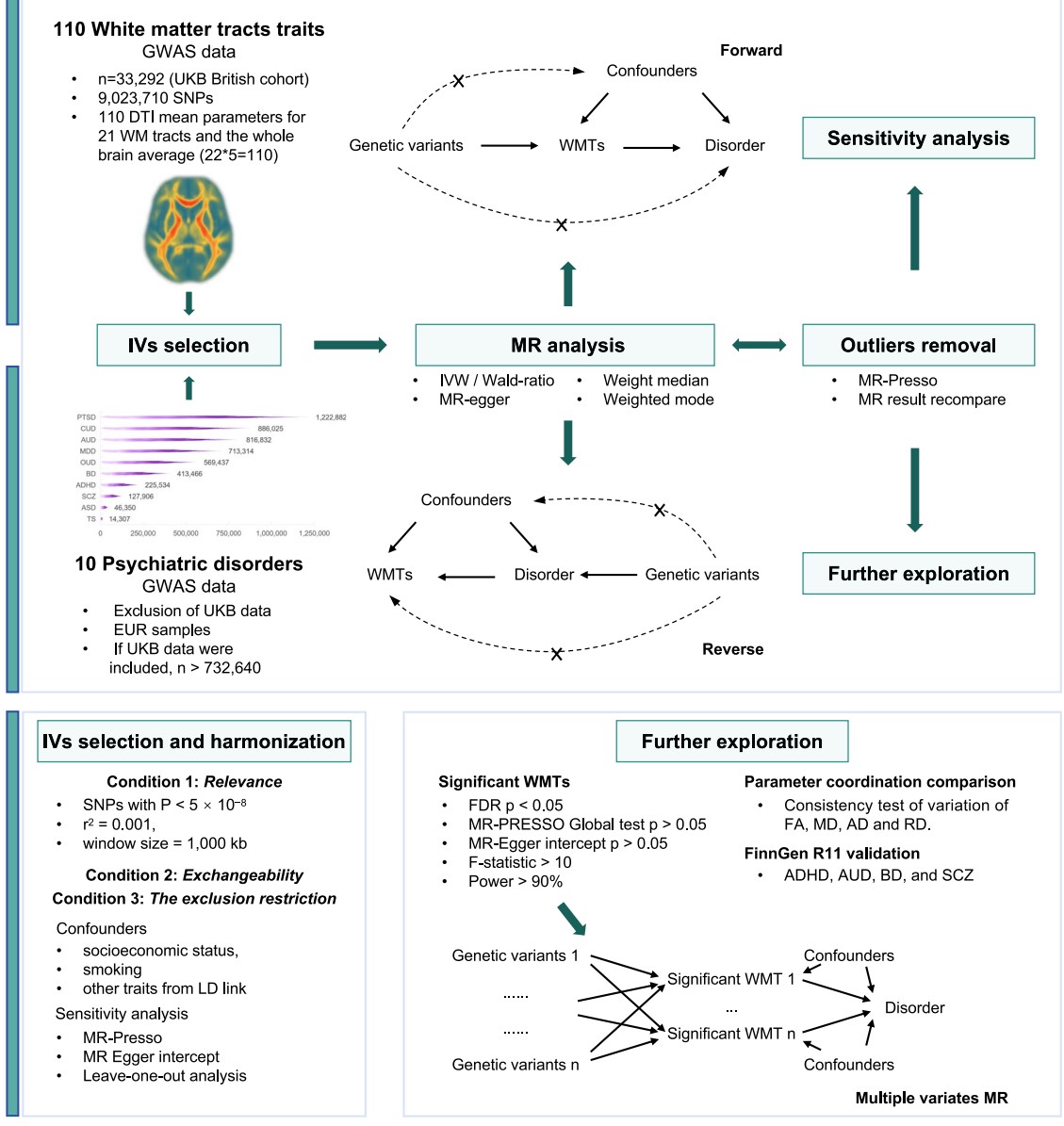

**Fig. 1 | Workflow of the causal inference between WMTs and psychiatric disorders.** AD axial diffusivities, ADHD attention deficit hyperactivity disorder, ASD autism disorder, AUD alcohol use disorder, BD bipolar disorder, CUD cannabis use disorder, FA fractional anisotropy, MD mean diffusivities, MDD major depression disorder, MO mode of anisotropy, MR mendelian randomization, PTSD post-traumatic stress disorder, RD radial diffusivities, SCZ schizophrenia, TS Tourette syndrome, WMTs white matter tracts, OUD opioid use disorder.

(UKB) data, with the sample overlap being <5%. The GWAS sample size of psychiatric disorders ranged from 14,307 to 1,222,882. Further descriptions and downloald links of GWAS data can be obtained in Supplementary Data 1 and Supplementary Data 2. All MR processes followed the Burgess[23] and Strengthening the Reporting of Observational Studies in Epidemiology guidelines[24] and the checklist was in Table S2.

Before addressing confounding factors, we screened 1667 single nucleotide polymorphism (SNP) loci as IVs for forward MR (Supplementary Data 3) and 301 for reverse MR (Supplementary Data 4), leaving 1376 and 301, respectively, after confounders removal (Supplementary Data 5–8). Besides previously reported confounders such as socioeconomic status and education, additional non-brain structural and nonpsychiatric factors were included, such as pulse pressure (rs2645466), coronary artery disease (rs4894803), and uterine leiomyoma or ER-positive breast cancer (rs10828248), etc. (Supplementary Data 7). After outliers detection (Supplementary Data 9–10), 11,745 SNPs pairs between WMTs and psychiatric disorders were validated for forward MR analysis (Supplementary Data 11),

and 30,343 SNPs pairs between psychiatric disorders and WMTs were validated for reverse MR analysis (Supplementary Data 12).

In the results of forward MR, 150 WMTs-psychiatric disorders pairs were nominally significant ($P_{raw}$ < 0.05). After conducting sensitivity analysis, limiting the number of SNPs and performing FDR correction, we identified significant associations between WMTs and: attention deficit hyperactivity disorder (ADHD) (1/108 tests), Tourette syndrome (TS) (1/108), BD (1/109), schizophrenia (SCZ) (3/109), MDD (1/109), opioid use disorder (OUD) (2/103), AUD (1/109), and PTSD (14/109). Additionally, we observed the presence of reverse causal associations between AUD and WMTs. The results and outlier analysis details of forward MR and reverse MR can be found in Supplementary Data 13–20. During statistical analysis, the $F$ values of the IVs in the final results were all >20, and the statistical power was >90%. All reported associations we reported below had passed FDR correction (FDR $P$ < 0.05; see Supplementary Data 13 and Supplementary Data 20 for full statistics). Finally, our findings indicate causal associations between WMTs and SCZ, BD, PTSD, MDD, AUD, OUD,

**Table 2 | Description of all GWAS summary-level data**

| | Source | Parameters | Number of tracts | Number of DTI traits | Sample (total) | Pubmed ID |
|---|---|---|---|---|---|---|
| **GWAS summary-level data of white matter tracts** | | | | | | |
| White matter tracts | Zhao et al.[36] | AD, FA, MD, MO, RD | 21 | 110 | 33,292 | 34140357 |
| **GWAS summary-level data of psychiatric disorders (Bidirectional MR)** | | | | | | |
| **Disease** | **Source** | **Sample (cases)** | **Sample (control)** | **Sample (total)** | **Ancestry** | **Pubmed ID** |
| Alcohol use disorder | | 113,325 | 639,923 | 753,248 | EUR | 38062264 |
| | iPSYCH | 3141 | 18,970 | 22,111 | EUR | |
| | MVP | 80,028 | 368,113 | 448,141 | EUR | |
| | UKB | - | - | - | EUR | |
| | FinnGen | 8866 | 209,926 | 218,792 | EUR | |
| | QIMR | 10,785 | 10,848 | 21,633 | EUR | |
| | PGC | 9938 | 30,992 | 40,930 | EUR | |
| | Yale-Penn 3 | 567 | 1074 | 1641 | EUR | |
| Attention deficit hyperactivity disorder | | 38,691 | 186,843 | 225,534 | EUR | 36702997 |
| | iPSYCH | - | - | - | EUR | |
| | deCODE | - | - | - | EUR | |
| | PGC | - | - | - | EUR | |
| Autism disorder | | 18,381 | 27,969 | 46,350 | EUR | 30804558 |
| | iPSYCH | 13,076 | 22,664 | 35,740 | EUR | |
| | PGC | 5305 | 5305 | 10,610 | EUR | |
| | FinnGen | - | - | - | EUR | |
| Bipolar disorder | PGC | 41,917 | 371,549 | 413,466 | EUR | 34002096 |
| Cannabis use disorder | | 42,281 | 843,744 | 886,025 | EUR | 37985822 |
| | PGC, deCODE | 14,522 | 298,941 | 313,463 | EUR | |
| | MVP | 22,260 | 423,587 | 445,847 | EUR | |
| | iPSYCH | 4733 | 95,657 | 100,390 | EUR | |
| | MGB | 456 | 24,088 | 24,544 | EUR | |
| | Yale-Penn 3 | 310 | 1471 | 1781 | EUR | |
| Major depression | | 184,270 | 529,044 | 713,314 | EUR | 37464041 |
| | iPSYCH | 29,158 | 38,142 | 67,300 | EUR | |
| | FinnGen | 28,098 | 228,817 | 256,915 | EUR | |
| | MVP | 83,810 | 166,405 | 250,215 | EUR | |
| | PGC | 43,204 | 95,680 | 138,884 | EUR | |
| Opioid Use Disorder | MVP, PGC, iPSYCH, FinnGen, Partners Biobank, BioVU, and Yale-Penn 3 | 15,251 | 554,186 | 569,437 | EUR | 35879402 |
| Post-traumatic stress disorder | PGC | 137,136 | 1,085,746 | 1,222,882 | EUR | 38637617 |
| Schizophrenia | PGC | 52,017 | 75,889 | 127,906 | EUR | 35396580 |
| Tourette syndrome | PGC | 4819 | 9488 | 14,307 | EUR | 30818990 |
| **GWAS summary-level data of psychiatric disorders (Replication)** | | | | | | |
| **Disease** | **Source** | **Sample (cases)** | **Sample (control)** | **Sample (total)** | **Ancestry** | **Pubmed ID** |
| Alcohol use disorder | FinnGen | 18,695 | 435,038 | 453,733 | EUR | - |
| Attention deficit hyperactivity disorder | FinnGen | 3702 | 445,327 | 449,029 | EUR | - |
| Bipolar disorder | FinnGen | 8209 | 394,756 | 402,965 | EUR | - |
| Schizophrenia | FinnGen | 6933 | 439,144 | 446,077 | EUR | - |

All download links were in Supplementary Data 2.
*AD* Axial diffusivities, *deCODE* deCODE Genetics, *DTI* Diffusion tensor imaging, *EUR* European, *FA* Fractional anisotropy, *iPSYCH* The Lundbeck Foundation Integrative Psychiatric Research, *MD* Mean diffusivities, *MGB* Mass General Brigham Biobank, *MVP* the Million Veteran Program, *MO* Mode of anisotropy, *PGC* Psychiatric Genomics Consortium, *QIMR* QIMR Berghofer Medical Research Institute, *RD* Radial diuffsivities, *UKB* UK Biobank.

ADHD, and TS and found reverse causal associations of AUD with four types of WMTs.

## Causal effects of white matter tracts on ADHD and TS

As shown in Fig. 2 and Supplementary Data 20, we uncovered causal associations of the column and body of fornix (FX) with ADHD, as well as of the fornix (cres)/stria terminalis (FXST) with TS. Specifically, for ADHD, a one s.d. increase in MO of the FX was associated with a 31·3% increase in the odds of ADHD (OR = 1·313, 95% CI = 1·117–1·543, $P_{raw} < 9·96 \times 10^{-4}$, FDR-$P < 4·89 \times 10^{-2}$). Conversely, for TS, a one s.d. increase in MO of the FXST was linked to a 71·4% decrease in the odds of TS (OR = 0·286, 95% CI = 0·166–0·493, $P_{raw} < 6·61 \times 10^{-6}$, FDR-$P < 5·44 \times 10^{-4}$).

## Causal effects of white matter tracts on BD and SCZ

As shown in Fig. 2 and Supplementary Data 20, we identified a causal association of FXST and BD, as well as of FXST, superior corona radiata (SCR), and retrolenticular part of internal capsule (RLIC) with SCZ. We found that a one s.d. increase in MO of the FXST was associated with a 33·4% decrease in odds of BD (OR = 0·666, 95% CI = 0·544–0·816, $P_{raw} < 8·63 \times 10^{-5}$, FDR-$P < 7·48 \times 10^{-3}$). Specifically, one s.d. increase in FA of the FXST was associated with a 37·6% decrease in SCZ risk (OR = 0·724, 95% CI = 0·618–0·849, $P_{raw} < 6·74 \times 10^{-5}$, FDR-$P < 5·24 \times 10^{-3}$). Conversely, one s.d. increase in FA of the SCR led to 16·0% elevation in SCZ risk (OR = 1·160, 95% CI = 1·058–1·272, $P_{raw} < 1·09 \times 10^{-4}$, FDR-$P < 3·72 \times 10^{-2}$). Additionally, one s.d. increase in MO of RLIC was linked to

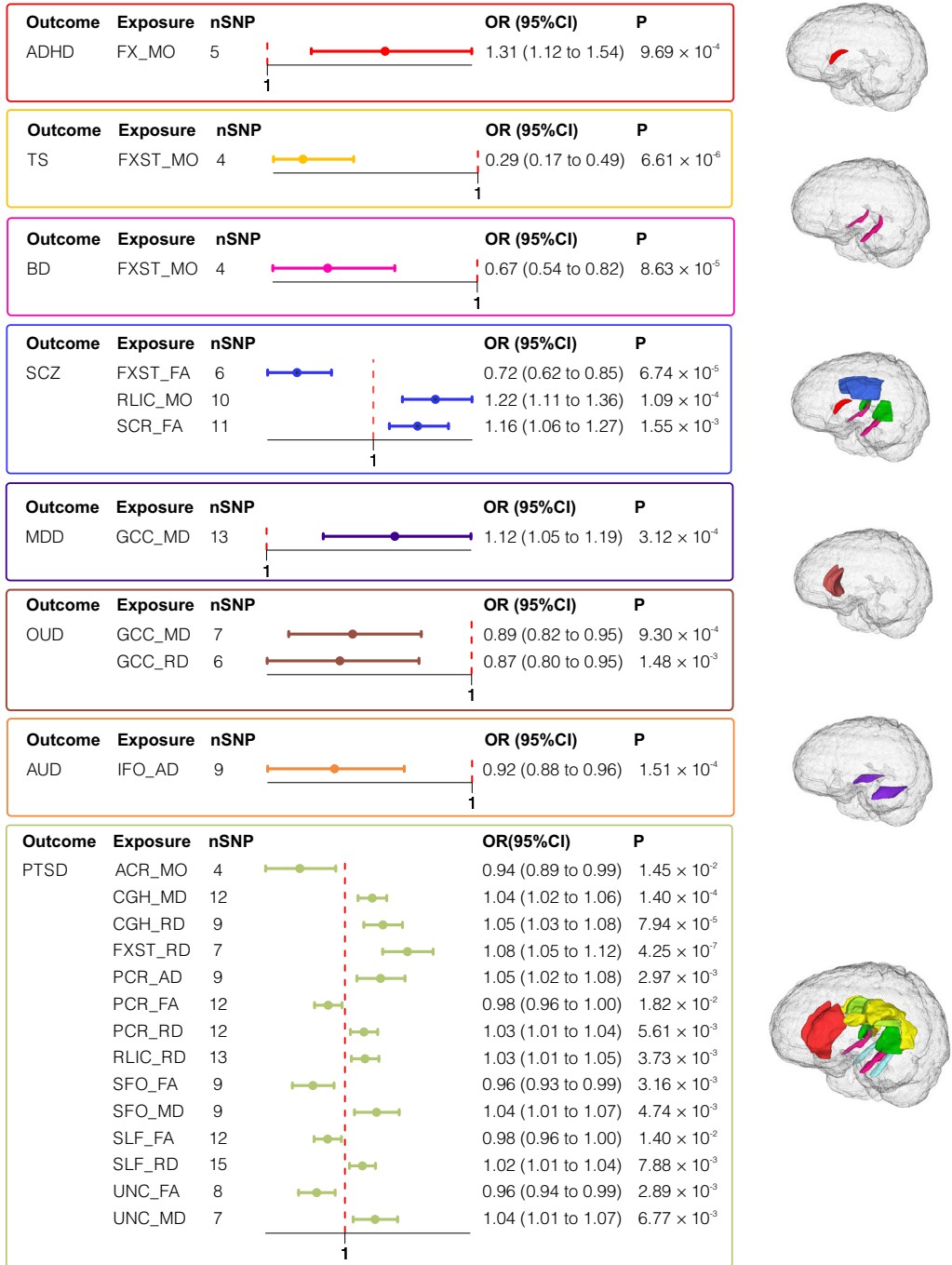

**Fig. 2 | Causalities in the forward MR.** The forest plot illustrates the significant causalities. The effect estimates displayed in the figure were calculated by IVW method, representing the OR of psychiatric disorders per 1 s.d. change in WMTs. And the error bars represent the 95% CI. Only FDR-significant (FDR p < 0.05) that passed heterogeneity and pleiotropytests, sensitivity analysis, and confounding factor adjustment were presented in the figure.

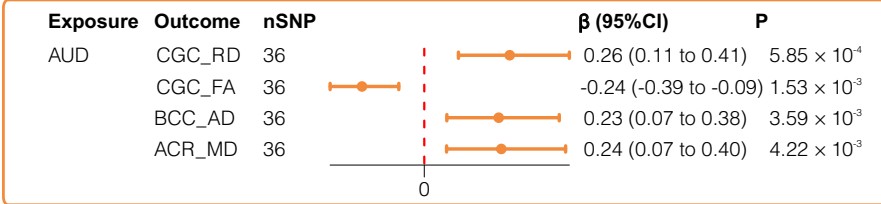

| Exposure | Outcome | nSNP | | β (95%CI) | P |
|---|---|---|---|---|---|
| AUD | CGC_RD | 36 | | 0.26 (0.11 to 0.41) | $5.85 \times 10^{-4}$ |
| | CGC_FA | 36 | | -0.24 (-0.39 to -0.09) | $1.53 \times 10^{-3}$ |
| | BCC_AD | 36 | | 0.23 (0.07 to 0.38) | $3.59 \times 10^{-3}$ |
| | ACR_MD | 36 | | 0.24 (0.07 to 0.40) | $4.22 \times 10^{-3}$ |

**Fig. 3 | Causalities in the reverse MR.** The forest plot illustrates the significant causalities. The effect estimates displayed in the figure were calculated by IVW method. And the error bars repersent the 95% CI. All statistical tests were two-sided, and only significant results (FDR-P < 0.05) that passed heterogeneity and pleiotropytests, sensitivity analysis, and confounding factor adjustment were presented in the figure.

a 22·5% (OR = 1·225, 95% CI = 1·105–1·357, $P_{raw} < 1·55 \times 10^{-3}$, FDR-$P < 5·24 \times 10^{-3}$) increase in odds of SCZ.

## Causal effects of white matter tracts on MDD, OUD, and AUD
As shown in Fig. 2 and Supplementary Data 20, we identified causal associations of the genu of corpus callosum (GCC) with MDD and OUD, as well as of the inferior fronto-occipital fasciculus (IFO) with AUD. For OUD, one s.d. increase in MD and RD of the GCC region was linked to 11·4% (OR = 0·886, 95% CI = 0·824–0·952, $P_{raw} < 9·30 \times 10^{-4}$, FDR-$P < 3·62 \times 10^{-2}$) and 12·7% (OR = 0·873, 95% CI = 0·803–0·949, $P_{raw} < 1·48 \times 10^{-3}$, FDR-$P < 3·62 \times 10^{-2}$) decreases in risk respectively. Similarly, one s.d. increase in MD was linked to 12·2% elevation in MDD risk (OR = 1·122, 95% CI = 1·054–1·195, $P_{raw} < 3·12 \times 10^{-4}$, FDR-$P < 3·27 \times 10^{-2}$). Furthermore, one s.d. increase in AD of the IFO was associated with a 7·9% decrease in AUD risk (OR = 0·921, 95% CI = 0·883–0·961, $P_{raw} < 1·51 \times 10^{-4}$, FDR-$P < 5·52 \times 10^{-3}$).

## Causal effects of white matter tracts on PTSD
As shown in Fig. 2 and Supplementary Data 20, we uncovered causal associations between PTSD and 7 WMTs: cingulum connecting to hippocampus (CGH), FXST, posterior corona radiata (PCR), RLIC, superior fronto-occipital fasciculus (SFO), superior longitudinal fasciculus (SLF) and uncinate fasciculus (UNC). One s.d. increase in FA for PCR, SFO, SLF, and UNC were found to decrease the risk of PTSD by 2·2% (OR = 0·978, 95% CI = 0·960–0·996, $P_{raw} < 1·82 \times 10^{-2}$, FDR-$P < 4·79 \times 10^{-3}$), 4·3% (OR = 0·957, 95% CI = 0·930–0·986, $P_{raw} < 3·16 \times 10^{-2}$, FDR-$P < 1·53 \times 10^{-2}$), 2·3% (OR = 0·977, 95% CI = 0·959–0·995, $P_{raw} < 1·40 \times 10^{-2}$, FDR-$P < 3·94 \times 10^{-3}$), and 3·8% (OR = 0·962, 95% CI = 0·938–0·987, $P_{raw} < 2·89 \times 10^{-3}$, FDR-$P < 1·47 \times 10^{-3}$), respectively. Furthermore, one s.d. increases in MD of CGH (OR = 1·036, 95% CI = 1·017–1·055, $P_{raw} < 1·40 \times 10^{-4}$, FDR-$P < 2·21 \times 10^{-3}$), SFO (OR = 1·042, 95% CI = 1·013–1·072, $P_{raw} < 4·74 \times 10^{-3}$, FDR-$P < 1·91 \times 10^{-2}$) and UNC (OR = 1·040, 95% CI = 1·011–1·070, $P_{raw} < 2·89 \times 10^{-3}$, FDR-$P < 2·36 \times 10^{-2}$), RD of CGH (OR = 1.050, 95% CI = 1·025–1·076, $P_{raw} < 7·94 \times 10^{-5}$, FDR-$P < 1·67 \times 10^{-3}$), FXST(OR = 1·083, 95% CI = 1·050–1·117, $P_{raw} < 4·25 \times 10^{-7}$, FDR-$P < 2·67 \times 10^{-5}$), PCR (OR = 1·025, 95% CI = 1·007–1·043, $P_{raw} < 2·08 \times 10^{-2}$), RLIC (OR = 1·027, 95% CI = 1·009–1·045, $P_{raw} < 3·73 \times 10^{-3}$, FDR-$P < 1·68 \times 10^{-2}$) and SLF (OR = 1·023, 95% CI = 1·006–1·041, $P_{raw} < 7·88 \times 10^{-3}$, FDR-$P < 2·61 \times 10^{-2}$), and AD of PCR (OR = 1·047, 95% CI = 1·016–1·080, $P_{raw} < 2·97 \times 10^{-3}$, FDR-$P < 1·49 \times 10^{-3}$) contributed to an increased risk of PTSD. Lastly, one s.d. increase in MO of the ACR was associated with a 6% decrease in the odds of PTSD (OR = 0·940, 95% CI = 0·895–0·988, $P_{raw} < 1·45 \times 10^{-2}$, FDR-$P < 4·03 \times 10^{-3}$).

## Reverse Mendelian randomization
As shown in Fig. 3. and Supplementary Data 20, we identified reverse causal associations of AUD with CGC, body of corpus callosum (BCC) and ACR. Higher risks of AUD was associated with decreased FA (IVW β = −0·246, 95% CI = −0·392 to −0·093, $P_{raw} < 1·53 \times 10^{-3}$, FDR-$P < 1·53 \times 10^{-3}$) and increased RD (IVW β = 0·259, 95% CI = 0·111–0·406, $P_{raw} < 5·84 \times 10^{-4}$, FDR-$P < 5·85 \times 10^{-4}$) of CGC, increased MD (IVW β = 0·236 95% CI = 0·074–0·398, $P_{raw} < 4·22 \times 10^{-3}$, FDR-$P < 4·22 \times 10^{-3}$) and increased AD (IVW β = 0.229, 95% CI = 0·075–0·383, $P_{uncorrected} < 3·60 \times 10^{-3}$, FDR-$P < 3·59 \times 10^{-3}$) in BCC.

## Multivariate Mendelian randomization and validation
We conducted multivariate MR analysis[25] on SCZ, OUD, and PTSD, and only the association between FXST_RD and PTSD (OR = 1·063, 95% CI = 1·004–1·125, $P < 3·61 \times 10^{-2}$) was retained. More details can be found in Supplementary Data 21. Subsequently, we partially validated our results using GWAS data for ADHD, AUD, BD, and SCZ from the FinnGen R11 database. The causal association between the MO of FX and ADHD in the forward MR analysis was replicated (OR = 1·806, 95% CI = 1·127–2·892, $P < 1·39 \times 10^{-2}$). Although BD did not yield identical replication results, it produced similar results: one s.d. increase in MO of FX (OR = 1·486, 95% CI = 1·065–2·072, $P < 1·97 \times 10^{-2}$) and RD of FXST (OR = 1·335, 95% CI = 1·025–1·740, $P < 3·21 \times 10^{-2}$) were associated with increasing risk of BD, and one s.d. increase FA of FXST (OR = 0·776, 95% CI = 0·613–0·982, $P < 3·50 \times 10^{-2}$) was associated with decreasing risk of BD. Taking into account the close spatial relationship (with a certain degree of overlap) between FX and FXST, as well as the potential correlations among the parameters FA, RD and MO, these similar results of BD could also be considered a successful replication.

## Confounding factors, outliers, and sensitivity analysis
To quantify confounding effects, we conducted parallel MR analyses using both pre-exclusion (N = 1667 SNPs) and post-exclusion (N = 1376 SNPs) IVs across all pairs. The comparative results (Supplementary Data 7 and Supplementary Data 16) revealed that only 2 associations exhibited meaningful changes of IVW estimates before/after confounder adjustment (among the 24 statistically significant associations): FX_MO-ADHD and GCC_MD-MDD transitioned from nonsignificant to significant (P > 0.05 → P < 0.05), while all others maintained consistent directionality and significance thresholds. Specifically, 1 SNP related to ascending thoracic aortic diameter was removed from FX_MO. And 6 SNPs, primarily associated with lipids, Alzheimer's disease, aging, and glutamic-oxaloacetic transaminase levels, were removed from the IVs of GCC_MD. Crucially, no associations showed β value reversals.

Regarding outliers, among the 24 significant results, 11 were obtained after excluding outliers: FXST_FA-SCZ (3), ACR_MO-PTSD (3), FXST_RD-PTSD (1), PCR_AD-PTSD (1), PCR_RD-PTSD (1), RLIC_RD-PTSD (2), SFO_MD-PTSD (3), SLF_FA-PTSD (4), SLF_RD-PTSD (4), UNC_FA-PTSD (3), and GCC_MD-MDD (3). Importantly, all IVW β values maintained directional consistency after outlier removal, indicating that the impact of outlier removal on the results is acceptable (please see the details in Supplementary Data 9 and Supplementary Data 17).

For sensitivity analyses, MR-PRESSO global tests and MR-Egger intercepts showed no evidence of pleiotropy (all P > 0.05). While we observed directional discrepancies between IVW and MR-Egger β values for FXST_FA-SCZ, the MR-Egger estimates were nonsignificant (all P > 0.05), and the direction of IVW β values were consistent with MR-median and MR-mode methods. Furthermore, among the 24 significant WMTs-psychiatric disorder associations, 16 showed evidence of sample overlap influence (p_difference < 0.05). Crucially, all associations maintained

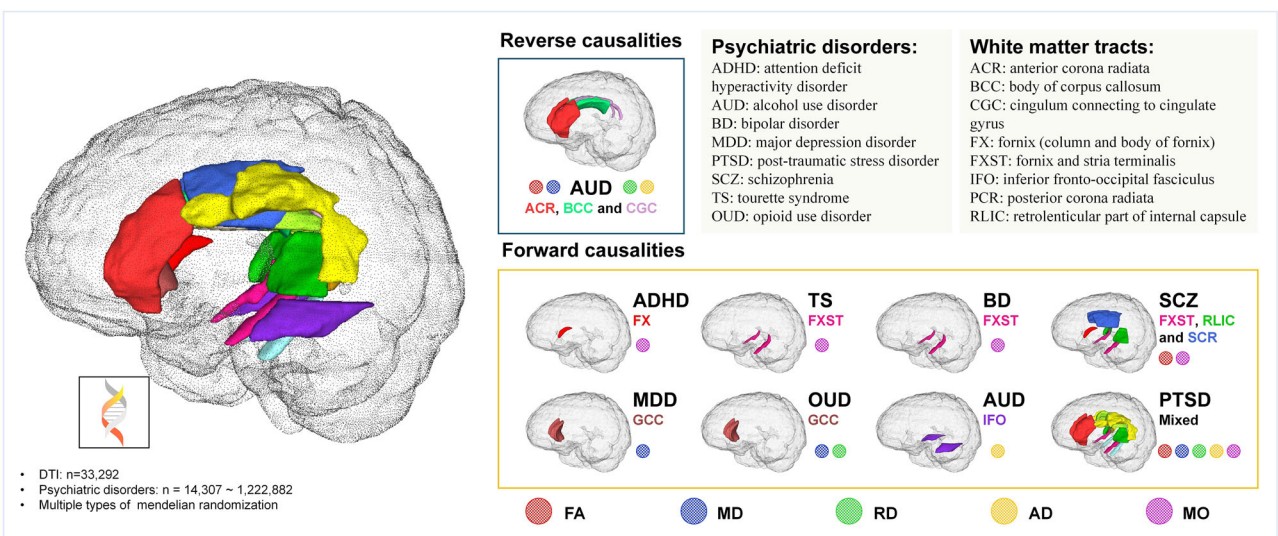

**Fig. 4 | Summary of results.** AD axial diffusivities, FA fractional anisotropy, MD mean diffusivities, MO diffusion tensor mode, RD radial diffusivities.

consistent $\beta$ value directionality before and after sample overlap adjustment (Supplementary Data 20). Notably, corrected effect estimates did not attenuate toward the null point; instead, effects strengthened through increased absolute $\beta$ values. This directional reinforcement aligns with and further substantiates our primary findings. Finally, we observed coordinated alterations across all DTI parameters. These findings collectively confirm the reliability of our MR causal inferences (please see the details in Supplementary Data 18).

## Discussion

To our knowledge, the current study represents the most comprehensive investigation to date of the causal associations between WMTs and psychiatric disorders. As summarized in Fig. 4, our findings not only establish forward causal associations between 24 WMTs-psychiatric disorder pairs but also reveal reverse causal associations between AUD and WMTs. These results indicate that WMTs may hold the potential to act as target regions for diagnosis or intervention in psychiatric disorders.

Recent MR studies have investigated the causal associations between imaging-derived phenotype (IDP) and psychiatric disorders, encompassing brain functional networks[19], cortical structures[21], white matter microstructure[22,26], and combinations of the three[19]. Building upon previous research, we have incorporated additional DTI parameters, expanded the scope to include more psychiatric disorders potentially related to WMTs, and significantly increased the sample sizes of psychiatric disorders. Guo et al.[18] utilized the GWAS data from Smith et al.[27] to validate the causal associations between IDP (which includes WMTs) and 10 psychiatric disorders, uncovering causal associations between WMTs and SCZ, as well as anorexia nervosa. Comparing with Guo et al.[18], we have incorporated additional DTI parameters (AD and RD), and expanded the scope to include more psychiatric disorders potentially related to WMTs (AUD, OUD and cannabis use disorder (CUD)). To minimize the potential bias, we excluded psychiatric disorders with a high risk of producing diagnosis bias based on our clinical experience. For example, generalized anxiety disorder and panic disorders are classified equally as anxious disorders in the database. But classifying them this way makes the different illnesses seem more similar than they really are, which makes it difficult for us to get interpretable results from data. Furthermore, we significantly increased the sample sizes (ADHD: from 53,293 to 225,534; BD: from 51,710 to 413,466; MDD: from 142,646 to 713,314; PTSD: from 146,660 to 1,222,882) of psychiatric disorders. Moreover, we implemented more specialized control measures to overcome limitations inherent in previous confounder and outlier removal methods. As previous GWAS studies had demonstrated, the genetic traits of brain structure were significantly stronger than those of brain functional

networks[27]. Therefore, compared to brain networks, utilizing SNPs related to brain structure would be more advantageous in revealing causal associations from a genetic perspective, yielding results with greater generalizability.

Although imaging indicators are indirect proxies, current studies tend to interpret decreases in FA and increases in MD as indicators of white matter myelin integrity damage[6,7] or axonal damage[8]. MO designates the type of anisotropy as a continuous measure, indicating differences in diffusion tensor shape ranging from planar (flattened cylinders) to linear (tubes)[7], which can help to map the direction of the WMTs. When dealing with more complex fiber structures where multiple fiber orientations coexist within a single voxel, techniques such as fiber orientation distribution are employed to measure this complexity. In such cases, MO can be utilized to signify the predominant or mean orientation when numerous fiber directions are identified. Although the association between WMTs and psychiatric disorders varies according to age, gender, medication, population (e.g., twins), disease severity stratification, and imaging control methods, as our mini review shown (Supplementary Materials), the preponderance of evidence currently supported that decrease in FA, increase in MD and RD, and alteration of AD were associated with the risk of psychiatric disorders and the severity of progression. Similarly, our MR results supported this tendency and aligned closely with 3 distinct patterns of neuropathology observed in the investigation of WMTs in patients[5]: (1) developmental abnormalities in limbic fibers (CGH, FX, and FXST), (2) abnormal maturation in long-range association fibers (IFO), and (3) severe developmental abnormalities and accelerated aging in callosal fibers (GCC and BCC).

The first main finding of this study is the causal associations between limbic system-related fibers (FXST or FX) and psychiatric disorders (SCZ, BD, ADHD, and TS). FXST, composed of FX and ST, is a fiber bundle associated with the limbic system[10]: the FX connects to the hippocampus, and the ST connects to the amygdala. Thus, alterations in FXST may contribute to emotional dysregulation[28,29] (in SCZ, BD, TS, related to the amygdala) or cognitive impairments[28,29] (in ADHD, related to the hippocampus). We speculated that the similar results of ADHD, TS, BD and SCZ are partly coming from the overlap of their similar pathogenesis, as suggested by evidence linking WMTs integrity in these regions to both genetic influences and environmental exposures, particularly in psychiatric developmental disorders like ADHD and autism spectrum disorder (ASD)[4,30], which could partly explain the similarity of some clinical symptom (such as mood instability). Furthermore, we notice that the significant results of FXST or FX are mostly related to the MO parameter, which potentially suggests that the loss of complexity in WMTs may be the underlying process

of psychiatric disorders. In conjunction with the existing evidence of close associations between the limbic cortical regions, limbic fibers and psychiatric disorders[10,28,29], we can conclude that both the cortex and WMTs of the limbic system are inextricably linked to the risk of ADHD, TS, BD, and SCZ.

The second main finding of this study is the results of PTSD and other psychiatric disorders. The results of PTSD suggested that it is an environmentally dependent psychiatric disorder with a relatively weaker association with genetics, which aligned with previous research[31–33]. Furthermore, we found a causal association between GCC and MDD, while GCC has connections with the frontal cortex and is involved in sensory and visuospatial processing[34,35], and it is closely related to social functioning impairment and dissociative symptoms[33]. Besides, the associations between WMTs and AUD in forward MR and reverse MR indicated that the substance itself (alcohol) is a significant risk factor in the occurrence of substance abuse disorders, but genetic factors are also influential.

This study had several limitations. Firstly, some overlap (<5%) was unavoidable in the samples, potentially introducing bias into our results. Although we used MRlap analysis to evaluate the effect of sample overlapping, MRlap could only evaluate the aggregate effects of sample overlap on results (indicating bias direction) without enabling precise quantification of bias effect size. Given that the MRlap findings ultimately aligned with our primary results, we conclude that while sample overlap introduced detectable bias, such bias would not alter the direction or attenuate the magnitude of our effect estimates. Secondly, we could not assess the impact of population and diagnostic stratification on the research results due to the use of publicly available GWAS databases. Thirdly, imaging data are indirect evidence, which limits our ability to infer the exact neurobiological processes from them. Fourthly, although we have controlled for both confounding factors and outliers, there could still be omitted factors and also potentially overall controlled factors (which are unnecessary and likely cause false negatives). Lastly, caution should be exercised when interpreting the clinical implications of OR values estimated by MR, as it utilizes risk SNP of exposure to explore the lifelong impact of exposures on outcomes, rather than the effects of specific interventions over a period of time, and cannot be equated with RCT studies.

In summary, by leveraging a genetically informed causal inference framework within the context of established (though often correlative) links between WMT microstructure and psychiatric disorders, our findings suggested potential causal associations between WMTs characteristics and psychiatric disorders through MR, which exhibited both similarities and differences—a pattern contingent upon the clinical feature similarities among different psychiatric disorders. This genetically informed approach suggests that specific WMTs characteristics could serve as hallmarks for psychiatric disorders[11], providing referential targeted regions for psychiatric disorders (diagnosis or intervention), thereby facilitating future clinical practice and scientific research.

## Methods
### Data sources
Data on white matter microstructure were obtained from Zhao et al.[36]. Zhao et al. conducted a GWAS on white matter microstructure using dMRI data from 33,292 individuals in the UKB British cohort, reporting a cumulative total of 9,023,710 SNPs (Chromosomes 1–22). The analysis method for white matter microstructure was derived from DTI models using the ENIGMA-DTI pipeline. They analyzed five primary DTI metrics (FA, MD, AD, RD, MO) across 21 brain WMTs, generating 110 DTI phenotypes for each individual. A detailed description of the GWAS can be found in Supplementary Data 1. The GWAS summary statistics are publicly available at Zenodo (https://zenodo.org/records/4549730), and the results can also be browsed through the BIG-KP knowledge portal (https://bigkp.org/).

We sourced data from public databases for psychiatric disorders, prioritizing GWAS-summary-level data when available. To minimize bias in results due to sample overlap between exposure and outcome, as well as ethnic differences, our selection criteria for psychiatric disorder GWAS data were as follows: (1) exclusion of UKB data; (2) inclusion of individuals of European descent (EUR); (3) if UKB data were included, the total sample size had to exceed 732,640 (ensuring a maximum sample overlap rate of <5%). According to Burgess et al., a 5% sample overlap in MR studies is estimated to introduce a bias of <0·15%[37]. The GWAS data on 10 psychiatric disorders, including ADHD[2], ASD[38], SCZ[39], BD[40], MDD[41], PTSD[42], AUD[43], OUD[44], CUD[45], and TS[46], were sourced from public databases. The GWAS sample size of psychiatric disorders ranged from 14,307 to 1,222,882.

### Missing data processing
We performed SNP position conversion based on chromosome and position for GWAS data lacking SNP ID using the GRCh37 reference. We matched the missing effective allele frequencies (EAF) using data from the author's tutorial or the 1000 Genomes Project. If EAF remained missing after this step, we substituted it with EAF = 0·5.

### Selection of instrumental variables (IVs)
MR analysis relies on three crucial assumptions: (1) the IVs should be associated with the exposure; (2) the IVs should be independent of confounding factors; (3) the IVs should only influence the outcome through the exposure directly. These assumptions necessitate a strong correlation between IVs and exposure and independence, and the pleiotropy of IVs does not interfere with the association between the exposure and the outcome[16]. The first two assumptions are primarily ensured through the selection of IVs. We selected SNPs with $P < 5 \times 10^{-8}$ to satisfy the strong correlation requirement. To ensure the independence of SNPs, we initially remove linkage disequilibrium (LD) due to close spatial associations, with parameters set as: $r^2 = 0.001$, window size = 1000 kb, gene reference = 1000 GENOME (EUR). Subsequently, we retrieved all traits corresponding to each SNP that passed both the P-value and LD screenings on LDlink (https://ldlink.nih.gov/?tab=home) and excluded those SNPs unrelated to brain imaging changes and psychiatric disorders, thereby controlling for confounding factors. These confounding factors include socioeconomic status, smoking, alcohol consumption, and many other factors that were not individually listed in previous MRs. Furthermore, to mitigate the impact of weak IVs, we calculated the F-statistic to measure the strength of IVs, with an F-statistic > 10 indicating a low risk of using weak instruments in MR analysis. Parameters calculated alongside the F-statistic included $R^2$ (the proportion of variance in the exposure explained by the genetic IVs), $n$ (the sample size of the GWAS for the exposure), and $k$ (the number of genetic IVs for the exposure).

### Bidirectional Mendelian randomization
Before conducting the MR analysis, we excluded variants with an EAF < 0·01. Furthermore, to ensure that the SNPs of the IVs originated from the same direction of the DNA strand and could be utilized in both the exposure and outcome datasets, we harmonized the exposure and outcome data and removed palindromic SNPs with EAF close to 0·5 (which could introduce potential strand flip problems).

For the forward MR, WMTs were used as exposure, and psychiatric disorders as outcome, while for the reverse MR, exposure and outcome were exchanged. The primary analytical method for MR was inverse variance weighted (IVW); supplementary analytical methods included Wald-ratio, MR weight median, MR Egger, and MR weighted mode. The statistical significance threshold for association was set at $P$ value < 0.05 with a false discovery rate (FDR) corrected using the IVW method. Specifically, FDR correction was applied per psychiatric disorder using the Benjamini–Hochberg procedure. For each disorder, we corrected for the number of tested WMT phenotypes. The odds ratio (OR) was used to represent the magnitude of the causal effect. Leveraging the multivariate regression concept in multivariate MR[25], we explored the presence of dominant traits in WMTs. To examine the generalization of the forward causal association between WMTs and psychiatric disorders, we revalidated the forward MR with the FinnGen R11 dataset[47].

## Sensitivity analysis and outlier screening

We employed MR-PRESSO, MR Egger intercept, Cochran's Q statistic[48,49], and leave-one-out analysis for sensitivity and pleiotropy assessments. And we used MRlap to assess the influence of sample overlap[50]. Initially, we used the MR-PRESSO global test to examine horizontal pleiotropy. The intercept of MR Egger represented the average pleiotropy of all IVs, with a value different from zero indicating the presence of directional pleiotropy. Heterogeneity was evaluated using Cochoran's Q. Lastly, we performed a "leave-one-out" analysis to test whether specific SNPs drove the causal association.

## Selection and interpretation of results

Consistent with Carter's recommendations for cautious interpretation of MR findings[51], we implemented a multi-layered framework to ensure biological plausibility alongside statistical significance.

First, all reported results satisfied strict statistical criteria: (1) FDR-adjusted p value < 0·05; (2) No evidence of pleiotropy (MR-PRESSO Global Test $p > 0·05$, MR-Egger intercept $p > 0·05$); and (3) Directional consistency across ≥3 of the 4 MR methods.

Second, we established neurobiological coherence criteria for DTI parameters. Given their known physiological interrelationships, causal associations were required to demonstrate directional consistency across metrics/parameters within each WMT. Specifically: (1) Decreased FA must correspond with either increased MD and/or increased RD and/or decreased AD; (2) Increased MD must align with either increased RD or decreased AD. This approach prioritized findings with greater biological plausibility by evaluating directional concordance through established neurobiological mechanisms. For example, reduced FA accompanied by increased MD and RD would consistently indicate myelin impairment. While statistically significant but biologically discordant results (e.g., isolated parameter changes lacking supporting directional patterns) would be dismissed, this tiered evaluation supplemented—rather than replaced—statistical significance thresholds.

Third, we implemented a parallel-instrument analytical approach to address potential overcorrection bias from indiscriminate exclusion of SNPs unrelated to brain imaging or psychiatric disorders. First, parallel MR analyses were conducted using both pre-exclusion and post-exclusion IVs sets. When exclusion of confounder-associated SNPs materially altered results, which was defined as both (1) reversal of $\beta$ value directionality and (2) transition across the statistical significance threshold ($P < 0.05$), we conducted a literature search to exam whether there are other studies that could support the alteration of results between the excluded traits and target outcomes.

## Statistics and reproducibility

All MR analyses were carried out in the R environment (version 4.4.1), using TwoSampleMR (version 0.6.5), MendelianRandomization (version 0.10.0) and Oneclick (version 5.1.10). All codes we used were provided at https://github.com/Yifan-xyy/Code_for_MR.git. In order to correct for multiple testing in performing forward MR and reverse MR analyses, the FDR method was used. The P value threshold for statistical significance was as follows: FDR-P < 0.05 for forward MR and reverse MR; 0.05 for multivariate MR and validation MR.

## Reporting summary

Further information on research design is available in the Nature Portfolio Reporting Summary linked to this article.

## Data availability

The datasets supporting the conclusions of this article are included within the article and its additional files. The specific data download links can be obtained from the original article or in Supplementary Data 1 and Supplementary Data 2.

## Code availability

All analyses were carried out in the R environment (version 4.4.1), using TwoSampleMR (version 0.6.5), MendelianRandomization (version 0.10.0) and Oneclick (version 5.1.10). The code utilized in this study can be obtained through email inquiry or download from link: https://github.com/Yifan-xyy/Code_for_MR.git.

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

## Acknowledgements

We appreciate the user-friendly and open-source code provided by the One Click Analyses team. We want to acknowledge the participants and investigators of the FinnGen R11 study. This work was supported by the National Key Research and Development Program of China (2023YFC2506800), the National Natural Science Foundation of China (no. 82371499) and Young Elite Scientists Sponsorship Program by CAST (2023QNRC001).

## Author contributions

Y.F. Yu, K. Yuan, and S.Z. Ai proposed the topic and main idea. Y.F. Yu, S.Z. Ai, and K. Yuan were responsible for data acquisition, analysis, or interpretation of data. Y.F. Yu wrote the initial draft. K. Yuan, S.Z. Ai, T.Y. Jia, X. Lin, S.H. Chang, Y.P. Bao, J. Sun, T. Gao, and J. Shi commented on and

revised the manuscript. All authors contributed to the final draft of the manuscript.

## Competing interests

The authors declare no competing interests.
