## [Transparent Peer Review file · Communications Biology]

Unveiling Causal Relationship between White Matter Tracts and Psychiatric Disorders

Corresponding Author: Professor Kai Yuan

Version 0:

Reviewer comments:

Reviewer #1

(Remarks to the Author)

The authors of Yu et al conducted Mendelian randomization (MR) analyses to better understand the causality between white matter tracts (WMTs) and psychiatric disorders. Biobank data and summary statistics from genome wide association studies were used in the study. They show that changes in WMTs can have an impact on the risk of these disorders. Those results are interesting, and provide additional insights into the genetic etiology of these disorders. However, I have some comments below.

In Section "Causal Effects ...", it is unclear why uncorrected p-values were used. It seems that the results are not highly significant for some disorders. The authors only described that their results are in Table S3 - Table S25, and the main results are in Fig. 2 and Table S22. Additional information, including adjusted p-values, test numbers, and the full results for Fig. 2, is needed.

The statement in the abstract is not very specific for disorders. Please provide additional details from the Results Section.

In Section "Overview of MR", please describe how many types of WMTs were tested for each disorder. Also, some information from the Method Section should be provided here.

The authors described that FDR p-values were used in their analyses. Please elaborate on the FDR approach.

Would it be possible for the authors to provide overlapping sample numbers?

Minor points.

In the Supplementary Tables, please add table numbers to sheets.

Reviewer #2

(Remarks to the Author)

Reviewer Comments

Overview

This manuscript explores the bidirectional causal relationships between various DTI parameters across 21 white matter tracts and ten psychiatric traits using Mendelian Randomization (MR). The authors report multiple putatively causal effects of 22 DTI phenotypes across eight psychiatric outcomes. Notably, alcohol use disorder (AUD) emerges as the only psychiatric trait demonstrating potential causal effects on four DTI measures. The study addresses a relevant and underexplored area in psychiatric genetics, poses meaningful questions, and presents compelling results. However, several major concerns must be addressed prior to acceptance.

1. Insufficient Reporting Detail in Results and Methods

This constitutes the most important weakness of this manuscript. Many parts of the Results and Methods require greater specificity, to better communicate the methods used and the study findings. As is, the text is difficult to follow and there are gaps that do not allow the reader to fully understand what was done in the study. The figures are very informative in terms of methods and results, but, similar detail is required in the text as well. Specifically:

-L128–135: Please ensure consistent reporting of odds ratios (ORs), confidence intervals (CIs), and p-values for all associations.

-L158–159: Expand on the multivariate MR findings. Include statistical results including effect sizes and p-values for all associations tested. Additionally, in this context it is unclear what it means that "only PTSD was retained". Please clarify.

-L160–164: Similarly, please provide detailed replication results, including effect sizes, CIs, and p-values for each tested association and region. Clarify the meaning of "biologically comparable meanings": what was tested to identify biological comparability? what were the results including metrics and p-values? Please also report which associations were found to be significant in the previous step but were deemed to not be "biologically meaningful" and why, providing the appropriate statistical results.

-L346–355: Please report all comparisons mentioned here explicitly and individually in the Results section, with their corresponding statistical results.

-L357–361: The sentence describing confounding factor assessment and manual descriptive interpretations is vague. Clarify whether these steps were part of the analysis pipeline or the discussion, and describe how empirical evidence was used for interpretation. Clearly state the number of tests performed per method for each trait pair. List all confounders identified and excluded, and explain how this influenced the results.

2. Sample Overlap and Bias Assessment

The manuscript states that datasets with less than 5% sample overlap were included, estimating a potential bias of <0.15%. However, this point needs further elaboration. The authors should:

-Discuss in which ways this overlap might still bias MR estimates (eg direction of bias).

-Include a sensitivity analysis using MRlap, an MR method that explicitly accounts for sample overlap, particularly for AUD and PTSD.

3. Mini Review Section (L166–175)

The purpose and placement of this section are unclear. It is not introduced as part of the introduction, and although search terms used are provided, there is no description of screening criteria, quality assessment, or study selection, making the reporting and methods used insufficient for a review. However, the authors did search the literature extensively, and it is worth reporting their findings outside the context of a "mini review". Thus, we recommend:

-Please remove Mini Review from the Results and integrate relevant insights into the Discussion. If deemed necessary, methodological details for the literature search conducted may be moved to the Supplementary Materials.

4. Appropriate citation of prior research

In L74, the authors claim that no association has been established between white matter tracts and psychiatric disorders in MR. However, Brainwide MR for anxiety disorders has yielded a positive association of white matter hyperintensities with anxiety disorders. Please refer to this in the introduction.

Minor Phrasing Issues

-L42: Rephrase to: "...which shows that eight psychiatric disorders are among the top 25 contributors to global disease burden."

-L74: Omit "either" from this sentence to help with clarity.

-L119: Remove the word "between".

-L198–199: Replace "anxious" with "generalized anxiety disorder" for clarity and precision.

Version 1:

Reviewer comments:

Reviewer #2

(Remarks to the Author)

Thank you for addressing those issues. You have clarified some of my concerns regarding the methods and I don't have any further questions

The Response to Reviewers

Causal Relationship between White Matter Tracts and Psychiatric Disorders: A Mendelian Randomization Study

Content

Reviewer #1	2
General Response to Reviewer #1	2
Point by Point Response to Reviewer #1	3
General Comment	3
Comment 1	4
Comment 2	6
Comment 3	7
Comment 4	9
Comment 5	10
Minor Comment	13
Reviewer #2	14
General Response to Reviewer #2	14
Point by Point Response to Reviewer #2	15
General Comment	15
Comment 1: Insufficient Reporting Detail in Results and Methods	16
Comment 2: Sample Overlap and Bias Assessment	27
Comment 3: Mini Review Section	30
Comment 4: Appropriate citation of prior research	31
Minor Phrasing Issues	32

Reviewer #1

General Response to Reviewer #1

Dear Reviewer #1,

We extend our sincere gratitude to you for your thorough evaluation and constructive critique of our manuscript. Your insightful comments have significantly enhanced the methodological rigor, analytical transparency, and scholarly presentation of this work. We have implemented all suggested revisions with careful consideration, as detailed in the point-by-point responses below.

Key improvements prompted by this review include:

1. Enhanced Statistical Reporting

- Comprehensive inclusion of both uncorrected (P_{raw}) and FDR-adjusted p-values (FDR-P) across all significant associations.
- Explicit documentation of test numbers per psychiatric disorder category.
- Expanded methodological details on FDR application (Benjamini-Hochberg procedure).
- Restructured "Overview of MR" section to integrate analytical workflow details

2. Improvement in the completeness of methodology and result reporting

- Added quantitative documentation of instrument variable processing stages (pre/post-confounder exclusion)
- Incorporated clinical specificity in abstract with exemplar effect sizes (PTSD, Tourette syndrome)

3. Clarification of Sample Overlap Bias and Supplementary Refinement

- Implemented MRlap analysis to quantify sample overlap bias
- Documented limitations of summary-level data constraints
- Standardized supplementary table labeling (S3-S25) with clear headers

These collective revisions—spanning Abstract, Results, Methods and Supplementary Materials—have strengthened causal inference validity while improving reader navigation through complex analytical pathways. We are indebted to you for your expertise in identifying these critical improvement opportunities, which have elevated the scientific quality and communicative clarity of this research. **For a detailed reply, please refer to the "point by point" response.**

Point by Point Response to Reviewer #1

Notes: Green boxes indicate the summary of revisions, grey boxes indicate revisions to the main text and abstract, yellow boxes indicate other edits (references, supplementary materials, etc.), **scarlet text represents text added** and ~~strikethrough marks~~ indicate deletions.

General Comment

" The authors of Yu et al conducted Mendelian randomization (MR) analyses to better understand the causality between white matter tracts (WMTs) and psychiatric disorders. Biobank data and summary statistics from genome wide association studies were used in the study. They show that changes in WMTs can have an impact on the risk of these disorders. Those results are interesting, and provide additional insights into the genetic etiology of these disorders. However, I have some comments below. "

Response:

We appreciate the reviewer's thoughtful comments and positive evaluation of our manuscript. We thank the reviewer for recognizing the value of our Biobank and GWAS-based approach in elucidating how WMT alterations impact psychiatric disorder risk. Regarding the subsequent comments, we will address each point thoroughly and revise the manuscript accordingly to strengthen our methodology and interpretation.

Comment 1

" In Section "Causal Effects ...", it is unclear why uncorrected p-values were used. It seems that the results are not highly significant for some disorders. The authors only described that their results are in Table S3 - Table S25, and the main results are in Fig. 2 and Table S22. Additional information, including adjusted p-values, test numbers, and the full results for Fig. 2, is needed. "

Response:

We apologize for the lack of clarity concerning the reporting of statistical significance. We wish to clarify that all findings reported in the **Results section**, including those presented in **Fig 2**, represent raw statistics that have passed the pre-specified FDR correction threshold for significance (FDR-P < 0.05). We agree that additional clarity was needed in the reporting of our statistical analyses and methodological workflow. Thus, based on your insightful comment:

- 1. We have revised our reporting strategy.**
- 2. We have incorporated explicit descriptions of both uncorrected and FDR-adjusted p-values, specified the number of tests performed per disorder.** While it is correct that some associations (notably for PTSD) yield uncorrected p-values closer to the FDR threshold and may not appear "highly significant" in magnitude, presenting the raw p-value and FDR-p together ensures greater transparency regarding the initial statistical strength.

The revised text is as follows (the scarlet text represents represent the text added):

Text Revision 1: Added a complete statistical report to **Results**

Causal effects of white matter tracts on ADHD and TS

(Clean version: page 6, line 124-128; Marked version: page 7, line 143-147)

... Specifically, for ADHD, a one s.d. increase in MO of the FX was associated with a 31.3% increase in the odds of ADHD (OR = 1.313, 95% CI = 1.117~1.543, $P_{\text{raw}} < 9.96 \times 10^{-4}$, $\text{FDR-P} < 4.89 \times 10^{-2}$). Conversely, for TS, a one s.d. increase in MO of the FXST was linked to a 71.4% decrease in the odds of TS (OR = 0.286, 95% CI = 0.166~0.493, $P_{\text{raw}} < 6.61 \times 10^{-6}$, $\text{FDR-P} < 5.44 \times 10^{-4}$).

Causal effects of white matter tracts on BD and SCZ

(Clean version: page 6, line 132-139; Marked version: page 8, line 151-158)

... We found that a one s.d. increase in MO of the FXST was associated with a 33.4% decrease in odds of BD (OR = 0.666, 95% CI = 0.544~0.816, $P_{\text{raw}} < 8.63 \times 10^{-5}$, $\text{FDR-P} < 7.48 \times 10^{-3}$). Specifically, one s.d. increase in FA of the FXST was associated with a 37.6% decrease in SCZ risk (OR = 0.724, 95% CI = 0.618~0.849, $P_{\text{raw}} < 6.74 \times 10^{-5}$, $\text{FDR-P} < 5.24 \times 10^{-3}$). Conversely, one s.d. increase in FA of the SCR led to 16.0% elevation in SCZ risk (OR = 1.160, 95% CI = 1.058~1.272, $P_{\text{raw}} < 1.09 \times 10^{-4}$, $\text{FDR-P} < 3.72 \times 10^{-2}$). Additionally, one s.d. increase in MO of RLIC was linked to a 22.5% (OR =

1.225, 95% CI = 1.105-1.357, $P_{\text{raw}} < 1.55 \times 10^{-3}$, $\text{FDR-P} < 5.24 \times 10^{-3}$) increase in odds of SCZ.

Causal effects of white matter tracts on MDD, OUD, and AUD

(Clean version: page 6, line 143-149; Marked version: page 8, line 162-169)

... For OUD, one s.d. increase in MD and RD of the GCC region was linked to 11.4% (OR = 0.886, 95% CI = 0.824~0.952, $P_{\text{raw}} < 9.30 \times 10^{-4}$, $\text{FDR-P} < 3.62 \times 10^{-2}$) and 12.7% (OR = 0.873, 95% CI = 0.803~0.949, $P_{\text{raw}} < 1.48 \times 10^{-3}$, $\text{FDR-P} < 3.62 \times 10^{-2}$) decreases in risk respectively. Similarly, one s.d. increase in MD was linked to 12.2% elevation in MDD risk (OR = 1.122, 95% CI = 1.054~1.195, $P_{\text{raw}} < 3.12 \times 10^{-4}$, $\text{FDR-P} < 3.27 \times 10^{-2}$). Furthermore, one s.d. increase in AD of the IFO was associated with a 7.9% decrease in AUD risk (OR = 0.921, 95% CI = 0.883~0.961, $P_{\text{raw}} < 1.51 \times 10^{-4}$, $\text{FDR-P} < 5.52 \times 10^{-3}$).

Causal effects of white matter tracts on PTSD

(Clean version: page 6, line 154-169; Marked version: page 8, line 174-190)

... One s.d. increase in FA for PCR, SFO, SLF, and UNC were found to decrease the risk of PTSD by 2.2% (OR = 0.978, 95% CI = 0.960 - 0.996, $P_{\text{raw}} < 1.82 \times 10^{-2}$, $\text{FDR-P} < 4.79 \times 10^{-3}$), 4.3% (OR = 0.957, 95% CI = 0.930~0.986, $P_{\text{raw}} < 3.16 \times 10^{-2}$, $\text{FDR-P} < 1.53 \times 10^{-2}$), 2.3% (OR = 0.977, 95% CI = 0.959~0.995, $P_{\text{raw}} < 1.40 \times 10^{-2}$, $\text{FDR-P} < 3.94 \times 10^{-3}$), and 3.8% (OR = 0.962, 95% CI = 0.938~0.987, $P_{\text{raw}} < 2.89 \times 10^{-3}$, $\text{FDR-P} < 1.47 \times 10^{-3}$), respectively. Furthermore, one s.d. increases in MD of CGH (OR = 1.036, 95% CI = 1.017~1.055, $P_{\text{raw}} < 1.40 \times 10^{-4}$, $\text{FDR-P} < 2.21 \times 10^{-3}$), SFO (OR = 1.042, 95% CI = 1.013~1.072, $P_{\text{raw}} < 4.74 \times 10^{-3}$, $\text{FDR-P} < 1.91 \times 10^{-2}$) and UNC (OR = 1.040, 95% CI = 1.011~1.070, $P_{\text{raw}} < 2.89 \times 10^{-3}$, $\text{FDR-P} < 2.36 \times 10^{-2}$), RD of CGH (OR = 1.050, 95% CI = 1.025~1.076, $P_{\text{raw}} < 7.94 \times 10^{-5}$, $\text{FDR-P} < 1.67 \times 10^{-3}$), FXST (OR = 1.083, 95% CI = 1.050~1.117, $P_{\text{raw}} < 4.25 \times 10^{-7}$, $\text{FDR-P} < 2.67 \times 10^{-5}$), PCR (OR = 1.025, 95% CI = 1.007~1.043, $P_{\text{raw}} < 2.08 \times 10^{-2}$), RLIC (OR = 1.027, 95% CI = 1.009~1.045, $P_{\text{raw}} < 3.73 \times 10^{-3}$, $\text{FDR-P} < 1.68 \times 10^{-2}$) and SLF (OR = 1.023, 95% CI = 1.006~1.041, $P_{\text{raw}} < 7.88 \times 10^{-3}$, $\text{FDR-P} < 2.61 \times 10^{-2}$), and AD of PCR (OR = 1.047, 95% CI = 1.016~1.080, $P_{\text{raw}} < 2.97 \times 10^{-3}$, $\text{FDR-P} < 1.49 \times 10^{-3}$) contributed to an increased risk of PTSD. Lastly, one s.d. increase in MO of the ACR was associated with a 6% decrease in the odds of PTSD (OR = 0.940, 95% CI = 0.895~0.988, $P_{\text{raw}} < 1.45 \times 10^{-2}$, $\text{FDR-P} < 4.03 \times 10^{-3}$).

Reverse Mendelian randomization

(Clean version: page 9, line 180-184; Marked version: page 11, line 203-207)

... Higher risks of AUD was associated with decreased FA (IVW β = -0.246, 95% CI = -0.392~-0.093, $P_{\text{raw}} < 1.53 \times 10^{-3}$, $\text{FDR-P} < 1.53 \times 10^{-3}$) and increased RD (IVW β = 0.259, 95% CI = 0.111~0.406, $P_{\text{raw}} < 5.84 \times 10^{-4}$, $\text{FDR-P} < 5.85 \times 10^{-4}$) of CGC, increased MD (IVW β = 0.236 95% CI = 0.074~0.398, $P_{\text{raw}} < 4.22 \times 10^{-3}$, $\text{FDR-P} < 4.22 \times 10^{-3}$) and increased AD (IVW β = 0.229, 95% CI = 0.075~0.383, $P_{\text{raw}} < 3.60 \times 10^{-3}$, $\text{FDR-P} < 3.59 \times 10^{-3}$) in BCC.

Comment 2

"The statement in the abstract is not very specific for disorders. Please provide additional details from the Results Section"

Response:

We sincerely thank the reviewer for highlighting the need for greater specificity regarding the psychiatric disorder findings in the **Abstract**. We agree that including concrete examples would strengthen the clarity and impact of our summary. To address this valuable feedback:

1. **We have revised the Abstract to explicitly name disorders cited in the Results Section and to incorporate representative effect sizes.** Importantly, we selected the disorder pairs with both the **strong statistical significance** (low P-values) and **large magnitude of effect** (most protective/risk-associated ORs), such as **PTSD and Tourette syndrome**, to illustrate the spectrum of our findings.
2. Considering the word limitation of the journal (< 150 words), we modified some **language expressions** in Abstract.

The revised text is as follows (the scarlet text represents represent the text added):

Text Revision 1: Added descriptions of representative results to **Abstract**

(Clean version: page 2, line 30-41; Marked version: page 3, line 33-53)

White matter tracts (WMTs), which mediate information transmission in the brain, are closely associated with the pathogenesis of psychiatric disorders, yet the causality of their associations remain unclear. Thus, we employed two-sample bidirectional Mendelian randomization to explore the causality between WMTs and 10 psychiatric disorders. **We found that one standard deviation changes of WMTs metrics modified risks for 8 psychiatric disorders by 2.2% to 71.4%. For example, increased fornix/stria terminalis radial diffusivity elevated PTSD risk by 8.3% (OR = 1.083, 95%CI = 1.050 - 1.117; FDR-P < 2.67 × 10⁻⁵), while heightened mode anisotropy reduced Tourette syndrome risk by 71.4% (OR = 0.286, 95%CI = 0.166 - 0.493; FDR-P < 5.44 × 10⁻⁴). Reversely, alcohol use disorder increased the risk of WMTs abnormalities.** Our study provides novel insights into the potential causality between WMTs and psychiatric disorders, indicating that alterations of WMTs may serve as biomarkers for psychiatric disorders.

Comment 3

"In Section "Overview of MR", please describe how many types of WMTs were tested for each disorder. Also, some information from the Method Section should be provided here."

Response:

We sincerely appreciate the reviewer's valuable comments, which we believe significantly contribute to improving the quality and clarity of our manuscript. We agree that additional clarity was needed in the reporting of our statistical analyses and methodological workflow. To address your comments comprehensively, we have implemented the following revisions:

- 1. Specification of WMTs Tested:** The "Overview of MR" section now clearly states the number of WMTs tested for association with each psychiatric disorder in the MR analysis.
- 2. Enhanced Result Details:** As suggested, we have integrated key methodological details (Data Sources, IV Selection, Sensitivity Analysis, Outlier Screening) from the Supplementary Methods section directly into the main text's "Overview of MR" section. We replaced the previous generic reference to "Table S3-S25" with a step-by-step documentation of the analytical workflow, sequentially referencing the relevant supplementary tables. This provides a clearer roadmap from data acquisition through processing to final results.

The revised text is as follows (the scarlet text represents represent the text added):

Text Revision 1: Added details of the results and integrate information from **Methods** to **Results** Overview of MR

(Clean version: page 4-5, line 87-113; Marked version: page 5-6, line 100-132)

The research design is illustrated in **Fig. 1.** and the baseline characteristics of genome-wide association studies (GWAS) are described in **Table 2.** Among all included GWAS data of psychiatric disorders, only AUD and PTSD included a small portion of UK Biobank data, with the sample overlap being < 5%. The GWAS sample size of psychiatric disorders ranged from 14,307 to 1,222,882. Further descriptions and download links of GWAS data can be obtained in **Table S3 and Table S4.** All Mendelian randomization (MR) processes followed the Burgess²³ and STROBE (Strengthening the Reporting of Observational Studies in Epidemiology) guidelines²⁴ and the checklist was in **Table S2.**

Before addressing confounding factors, we screened 1,667 SNP loci as instrumental variables for forward MR (**Table S5**) and 301 for reverse MR (**Table S6**), leaving 1,376 and 301 respectively after confounders removal (**Table S7 - S10**). Besides previously reported confounders such as socioeconomic status and education, additional non-brain structural and non-psychiatric factors were included, such as pulse pressure (rs2645466), coronary artery disease (rs4894803), and uterine leiomyoma or ER-positive breast cancer (rs10828248), etc. (**Table S9**). After outliers detection (**Table S11 - S12**), 11,745 SNPs pairs between WMTs and psychiatric disorders were validated for forward MR analysis (**Table S13**), and 30,343 SNPs pairs

between psychiatric disorders and WMTs were validated for reverse MR analysis (**Table S14**).

In the results of forward MR, 150 WMTs-psychiatric disorders pairs were nominally significant ($P_{\text{raw}} < 0.05$). After conducting sensitivity analysis, limiting the number of SNPs and performing FDR correction, we identified significant associations between WMTs and: ADHD (1/108 tests), TS (1/108), BD (1/109), SCZ (3/109), MDD (1/109), OUD (2/103), AUD (1/109), and PTSD (14/109). Additionally, we observed the presence of reverse causal associations between AUD and WMTs. The results and outlier analysis details of forward MR and reverse MR can be found in **Table S15 - S22**. During statistical analysis, the F-values of the instrumental variables in the final results were all > 20 , and the statistical power was $> 90\%$. All reported associations we reported below had passed FDR correction ($\text{FDR } P < 0.05$; see **Table S15 and Table S22** for full statistics). Finally, our findings indicate causal associations between WMTs and SCZ, BD, PTSD, MDD, AUD, OUD, ADHD, and TS and found reverse causal associations of AUD with 4 types of WMTs.

Comment 4

" The authors described that FDR p-values were used in their analyses. Please elaborate on the FDR approach. "

Response:

We sincerely appreciate the reviewer's insightful question regarding our application of FDR correction. Your query has prompted us to significantly enhance the methodological transparency of our statistical approach, and we provide the following detailed clarification through targeted manuscript revisions:

- 1. Dual P-Value Reporting for Transparency:** As noted in our response to **Comment #1**, all key findings in the main text now display both: Raw P-values and FDR-P values.
- 2. Clarification of test numbers of each disorder in Results:** As noted in our response to **Comment #3** we now explicitly state the number of how many types of WMTs were tested for each disorder.
- 3. Expanded Methodological Details:** In *Methods* → "**Bidirectional Mendelian Randomization**", we have expanded the description of our FDR correction approach.

The revised text is as follows (the scarlet text represents represent the text added):

The **revised text of P-value reporting and test numbers** have been noted in the text of **Comment #1 and Comment #3** in the section "Overview of MR" and "Causal effects of". Please check it.

Text Revision 1: Added methodological details of FDR to **Methods**

Bidirectional Mendelian randomization

(Clean version: page 16, line 392-394; Marked version: page 18, line 427-429)

"... The statistical significance threshold for association was set at $P\text{-value} < 0.05$ with a false discovery rate (FDR) corrected using the IVW method. **Specifically, FDR correction was applied per psychiatric disorder using the Benjamini-Hochberg procedure. For each disorder, we corrected for the number of tested WMT phenotypes ...**"

Comment 5

"Would it be possible for the authors to provide overlapping sample numbers?"

Response:

We sincerely thank the reviewer for raising this important methodological consideration regarding sample overlap quantification. We fully agree that transparency in reporting potential sample overlap is critical for robust Mendelian randomization analysis. While we recognize the value of providing exact overlapping participant counts, this is unfortunately not feasible in our study due to our exclusive reliance on publicly available summary-level GWAS data (individual-level data were inaccessible). To address this limitation comprehensively, we implemented a three-tiered approach:

1. Proactive Overlap Minimization:

As detailed in our *Methods - Data Source* section: "To minimize bias in results due to sample overlap between exposure and outcome, as well as ethnic differences, our selection criteria for psychiatric disorder GWAS data were as follows: (1) exclusion of UKB data; (2) inclusion of individuals of European descent (EUR); (3) if UKB data were included, the total sample size had to exceed 732,640 (ensuring a maximum sample overlap rate of <5%). According to Burgess et al., a 5% sample overlap in MR studies is estimated to introduce a bias of <0.15%".

2. Comprehensive Data Documentation: Complete cohort sources, sample sizes and data download links for both WMTs and psychiatric disorders are provided in **Table S3 and Table S4**.

3. Quantitative Overlap Bias Assessment: To directly address the reviewer's concern, we implemented MRlap analysis (Mounier & Kutalik, 2023)

(1) Key Findings from MRlap Analysis: Among the 24 significant WMTs-psychiatric disorder associations identified in our primary analysis:

- 16 demonstrated detectable sample overlap influence (P difference < 0.05)
- Crucially, all 24 associations maintained consistent effect directionality after bias adjustment
- Notably, corrected effect estimates consistently strengthened rather than attenuated toward null (absolute β -values increased)

(2) Technical Note: As MRlap requires raw summary statistics without confounder/outlier filtering, the IV sets differ from our primary analysis. While this precludes direct effect-size comparisons, the complete directional concordance between MRlap-adjusted and primary

results provides robust validation against sample overlap bias. We have transparently documented this limitation in our *Discussion* section.

(3) Results of MRlap:

Exposure	Outcome	observed_effect	corrected_effect	p_difference
FX_MO	ADHD	0.05	0.07	5.57E-02
IFO_AD	AUD	(0.09)	(0.13)	7.76E-08
FXST_MO	BD	(0.08)	(0.11)	1.03E-05
GCC_MD	MDD	0.02	0.02	5.49E-02
GCC_MD	ODD	(0.08)	(0.10)	8.23E-03
GCC_RD	ODD	(0.06)	(0.09)	6.21E-02
ACR_MO	PTSD	(0.02)	(0.03)	3.82E-01
CGH_MD	PTSD	0.02	0.03	1.83E-02
CGH_RD	PTSD	0.02	0.03	3.06E-02
FXST_RD	PTSD	0.04	0.06	1.12E-02
PCR_AD	PTSD	0.04	0.06	1.29E-05
PCR_FA	PTSD	(0.02)	(0.03)	1.36E-03
PCR_RD	PTSD	0.03	0.04	1.18E-02
RLIC_RD	PTSD	0.04	0.05	6.48E-05
SFO_FA	PTSD	(0.04)	(0.05)	5.56E-04
SFO_MD	PTSD	0.02	0.03	6.15E-02
SLF_FA	PTSD	(0.02)	(0.03)	3.27E-02
SLF_RD	PTSD	0.02	0.02	3.81E-02
UNC_FA	PTSD	(0.02)	(0.03)	1.75E-01
UNC_MD	PTSD	0.02	0.03	1.07E-01
FXST_FA	SCZ	(0.05)	(0.07)	3.54E-01
RLIC_MO	SCZ	0.07	0.10	2.25E-04
SCR_FA	SCZ	0.05	0.07	2.94E-02
FXST_MO	TS	(0.51)	(0.75)	5.70E-08

The revised text is as follows (the scarlet text represents represent the text added):

Text Revision 1: Added explicit description of MRlap and interpretation framework to *Methods*

Sensitivity analysis and outlier screening

(Clean version: page 16, line 401-402; Marked version: page 19, line 437)

We employed MR-PRESSO, MR Egger intercept, Cochran's Q statistic^{48,49}, and leave-one-out analysis for sensitivity and pleiotropy assessments. And we used MRlap to assess the influence of sample overlap

⁵⁰.

Text Revision 2: Incorporated key findings with directional concordance emphasis to *Results*

Confounding factors, outliers, and sensitivity analysis

(Clean version: page 10, line 227-232; Marked version: page 12-13, line 252-257)

... Furthermore, among the 24 significant WMTs-psychiatric disorder associations, 16 showed evidence of sample overlap influence ($p_{\text{difference}} < 0.05$). Crucially, all associations maintained consistent β -value directionality before and after sample overlap adjustment (**Table S22**). Notably, corrected effect estimates did not attenuate toward the null point; instead, effects strengthened through increased absolute β -values. This directional reinforcement aligns with and further substantiates our primary findings. Finally, we observed coordinated alterations across all diffusion tensor imaging parameters. These findings collectively confirm the reliability of our MR causal inferences (please see the details in **Table S20**).

Text Revision 3: Added nuanced interpretation of MRlap results acknowledging technical constraints while highlighting confirmatory value to *Discussion*

(Clean version: page 13-14, line 310-315; Marked version: page 16, line 345-350)

This study had several limitations. Firstly, some overlap (<5%) was unavoidable in the samples, potentially introducing bias into our results. Although we used MRlap analysis to evaluate the effect of sample overlapping, MRlap could only evaluate the aggregate effects of sample overlap on results (indicating bias direction) without enabling precise quantification of bias effect size. Given that the MRlap findings ultimately aligned with our primary results, we conclude that while sample overlap introduced detectable bias, such bias would not alter the direction or attenuate the magnitude of our effect estimates ...

Supplementary Table Revision: The results of MRlap have been added in Table S22.

Added new reference:

Mounier, N. & Kutalik, Z. Bias correction for inverse variance weighting Mendelian randomization. *Genet Epidemiol* **47**, 314-331 (2023).

Minor Comment

"In the Supplementary Tables, please add table numbers to sheets."

Response:

We sincerely appreciate the reviewer's meticulous attention to detail regarding supplementary material organization. Thank you for highlighting this important formatting consideration. We have implemented the following improvements across all supplementary tables:

- 1. Clearly displayed table numbers (S3–S25) on every sheet**
- 2. Added standardized titles in the header row of each worksheet**
- 3. Verified consistent labeling throughout all supplementary files**

We're happy to provide the updated supplementary files for verification and welcome any additional formatting suggestions to enhance accessibility.

Reviewer #2

General Response to Reviewer #2

Dear Reviewer #2,

We sincerely thank you for your exceptionally thorough and constructive critique, which has been instrumental in strengthening our manuscript. We have implemented all recommendations with maximal rigor, prioritizing enhanced methodological transparency, biological plausibility, and statistical robustness. Your expertise guided critical enhancements across 3 key dimensions:

1. Enhanced Statistical & Methodological Rigor

- Comprehensive reporting of all associations with exact p-values (both P_{raw} and FDR-P), OR, and 95% CIs throughout Results.
- Replacement of "biologically comparable meanings" with a formalized directional concordance validation framework defining neurobiological consistency criteria for DTI parameters.
- Full documentation of IV selection, parallel MR analyses, and pleiotropy controls.

2. Explicit Bias Quantification

- MRlap implementation for all 24 significant associations, confirming consistent effect directionality despite sample overlap.
- Refine the process of assessing confounding factors.

3. Structural & Supplemental Optimization

- Integration of the "mini-review" into Discussion with supplemental methodology.
- Standardized supplementary labeling (Tables S3-S25) and expanded sensitivity reporting.

We are deeply indebted for your rigorous engagement—every recommendation elevated causal inference validity while exemplifying peer review at its most constructive. All changes are explicitly marked (underlined) for your review, with full technical details in our point-by-point reply.

Point by Point Response to Reviewer #2

Notes: Green boxes indicate the summary of revisions, grey boxes indicate revisions to the main text and abstract, yellow boxes indicate other edits (references, supplementary materials, etc.), **scarlet text represents text additions**, and ~~strikethrough marks~~ indicate deletions.

General Comment

" This manuscript explores the bidirectional causal relationships between various DTI parameters across 21 white matter tracts and ten psychiatric traits using Mendelian Randomization (MR). The authors report multiple putatively causal effects of 22 DTI phenotypes across eight psychiatric outcomes. Notably, alcohol use disorder (AUD) emerges as the only psychiatric trait demonstrating potential causal effects on four DTI measures. The study addresses a relevant and underexplored area in psychiatric genetics, poses meaningful questions, and presents compelling results. However, several major concerns must be addressed prior to acceptance. "

Response:

Thanks for your thoughtful assessment of our manuscript. We greatly appreciate your recognition of the study's relevance in exploring bidirectional causal relationships between DTI-derived white matter tract phenotypes and ten psychiatric traits via Mendelian Randomization (MR). We are particularly encouraged by the reviewer's acknowledgment of the our results.

We acknowledge that several major issues require attention and are fully prepared to address these comprehensively. We will carefully review each point, conduct additional analyses where necessary, and rigorously revise the manuscript to strengthen methodology, interpretation, and clarity. Detailed responses to each concern have been provided in the point-by-point response. Please check it, thanks again.

Comment 1: Insufficient Reporting Detail in Results and Methods

"This constitutes the most important weakness of this manuscript. Many parts of the Results and Methods require greater specificity, to better communicate the methods used and the study findings. As is, the text is difficult to follow and there are gaps that do not allow the reader to fully understand what was done in the study. The figures are very informative in terms of methods and results, but, similar detail is required in the text as well. Specifically:

Comment 1.1

-L128–135: Please ensure consistent reporting of odds ratios (ORs), confidence intervals (CIs), and p-values for all associations.

Response:

We sincerely thank the reviewer for highlighting the need for standardized reporting of statistical results. We appreciate this opportunity to enhance methodological rigor in our manuscript. We have now comprehensively addressed the suggestion through two key actions:

1. **Full Integration of PTSD Results:** Although initial selective reporting of PTSD associations aimed to maintain readability, we have now incorporated all 14 associations in the main text. This supersedes and replaces the **Supplemental Material** previously containing these results.
2. **Unified Statistical Reporting:** All significant associations (including ADHD, TS, BD, SCZ, MDD, OUD, AUD, PTSD and reverse MR) now consistently report: odds ratios (OR) with 95% confidence intervals (95% CI), raw p-values (P_{raw}), FDR-corrected p-values (FDR-P)

The revised text is as follows (**the scarlet text represents represent the text added**):

Text Revision 1: Added a complete statistical report to **Results**

Causal effects of white matter tracts on ADHD and TS

(Clean version: page 6, line 124-128; Marked version: page 7, line 143-147)

... Specifically, for ADHD, a one s.d. increase in MO of the FX was associated with a 31·3% increase in the odds of ADHD (OR = 1·313, 95% CI = 1·117~1·543, $P_{\text{raw}} < 9·96 \times 10^{-4}$, FDR-P < $4·89 \times 10^{-2}$). Conversely, for TS, a one s.d. increase in MO of the FXST was linked to a 71·4% decrease in the odds of TS (OR = 0·286, 95% CI = 0·166~0·493, $P_{\text{raw}} < 6·61 \times 10^{-6}$, FDR-P < $5·44 \times 10^{-4}$).

Causal effects of white matter tracts on BD and SCZ

(Clean version: page 6, line 132-139; Marked version: page 8, line 151-158)

... We found that a one s.d. increase in MO of the FXST was associated with a 33·4% decrease in odds of BD (OR = 0·666, 95% CI = 0·544~0·816, $P_{\text{raw}} < 8·63 \times 10^{-5}$, FDR-P < $7·48 \times 10^{-3}$). Specifically, one s.d. increase in FA of the FXST was associated with a 37·6% decrease in SCZ risk (OR = 0·724, 95%

CI = 0.618~0.849, $P_{\text{raw}} < 6.74 \times 10^{-5}$, $\text{FDR-P} < 5.24 \times 10^{-3}$). Conversely, one s.d. increase in FA of the SCR led to 16.0% elevation in SCZ risk (OR = 1.160, 95% CI = 1.058~1.272, $P_{\text{raw}} < 1.09 \times 10^{-4}$, $\text{FDR-P} < 3.72 \times 10^{-2}$). Additionally, one s.d. increase in MO of RLIC was linked to a 22.5% (OR = 1.225, 95% CI = 1.105~1.357, $P_{\text{raw}} < 1.55 \times 10^{-3}$, $\text{FDR-P} < 5.24 \times 10^{-3}$) increase in odds of SCZ.

Causal effects of white matter tracts on MDD, OUD, and AUD

(Clean version: page 6, line 143-149; Marked version: page 8, line 162-169)

... For OUD, one s.d. increase in MD and RD of the GCC region was linked to 11.4% (OR = 0.886, 95% CI = 0.824~0.952, $P_{\text{raw}} < 9.30 \times 10^{-4}$, $\text{FDR-P} < 3.62 \times 10^{-2}$) and 12.7% (OR = 0.873, 95% CI = 0.803~0.949, $P_{\text{raw}} < 1.48 \times 10^{-3}$, $\text{FDR-P} < 3.62 \times 10^{-2}$) decreases in risk respectively. Similarly, one s.d. increase in MD was linked to 12.2% elevation in MDD risk (OR = 1.122, 95% CI = 1.054~1.195, $P_{\text{raw}} < 3.12 \times 10^{-4}$, $\text{FDR-P} < 3.27 \times 10^{-2}$). Furthermore, one s.d. increase in AD of the IFO was associated with a 7.9% decrease in AUD risk (OR = 0.921, 95% CI = 0.883~0.961, $P_{\text{raw}} < 1.51 \times 10^{-4}$, $\text{FDR-P} < 5.52 \times 10^{-3}$).

Causal effects of white matter tracts on PTSD

(Clean version: page 6, line 154-169; Marked version: page 8, line 174-190)

... One s.d. increase in FA for PCR, SFO, SLF, and UNC were found to decrease the risk of PTSD by 2.2% (OR = 0.978, 95% CI = 0.960 - 0.996, $P_{\text{raw}} < 1.82 \times 10^{-2}$, $\text{FDR-P} < 4.79 \times 10^{-3}$), 4.3% (OR = 0.957, 95% CI = 0.930~0.986, $P_{\text{raw}} < 3.16 \times 10^{-2}$, $\text{FDR-P} < 1.53 \times 10^{-2}$), 2.3% (OR = 0.977, 95% CI = 0.959~0.995, $P_{\text{raw}} < 1.40 \times 10^{-2}$, $\text{FDR-P} < 3.94 \times 10^{-3}$), and 3.8% (OR = 0.962, 95% CI = 0.938~0.987, $P_{\text{raw}} < 2.89 \times 10^{-3}$, $\text{FDR-P} < 1.47 \times 10^{-3}$), respectively. Furthermore, one s.d. increases in MD of CGH (OR = 1.036, 95% CI = 1.017~1.055, $P_{\text{raw}} < 1.40 \times 10^{-4}$, $\text{FDR-P} < 2.21 \times 10^{-3}$), SFO (OR = 1.042, 95% CI = 1.013~1.072, $P_{\text{raw}} < 4.74 \times 10^{-3}$, $\text{FDR-P} < 1.91 \times 10^{-2}$) and UNC (OR = 1.040, 95% CI = 1.011~1.070, $P_{\text{raw}} < 2.89 \times 10^{-3}$, $\text{FDR-P} < 2.36 \times 10^{-2}$), RD of CGH (OR = 1.050, 95% CI = 1.025~1.076, $P_{\text{raw}} < 7.94 \times 10^{-5}$, $\text{FDR-P} < 1.67 \times 10^{-3}$), FXST (OR = 1.083, 95% CI = 1.050~1.117, $P_{\text{raw}} < 4.25 \times 10^{-7}$, $\text{FDR-P} < 2.67 \times 10^{-5}$), PCR (OR = 1.025, 95% CI = 1.007~1.043, $P_{\text{raw}} < 2.08 \times 10^{-2}$), RLIC (OR = 1.027, 95% CI = 1.009~1.045, $P_{\text{raw}} < 3.73 \times 10^{-3}$, $\text{FDR-P} < 1.68 \times 10^{-2}$) and SLF (OR = 1.023, 95% CI = 1.006~1.041, $P_{\text{raw}} < 7.88 \times 10^{-3}$, $\text{FDR-P} < 2.61 \times 10^{-2}$), and AD of PCR (OR = 1.047, 95% CI = 1.016~1.080, $P_{\text{raw}} < 2.97 \times 10^{-3}$, $\text{FDR-P} < 1.49 \times 10^{-3}$) contributed to an increased risk of PTSD. Lastly, one s.d. increase in MO of the ACR was associated with a 6% decrease in the odds of PTSD (OR = 0.940, 95% CI = 0.895~0.988, $P_{\text{raw}} < 1.45 \times 10^{-2}$, $\text{FDR-P} < 4.03 \times 10^{-3}$).

Reverse Mendelian randomization

(Clean version: page 9, line 180-184; Marked version: page 11, line 203-207)

... Higher risks of AUD was associated with decreased FA (IVW β = -0.246, 95% CI = -0.392~-0.093, $P_{\text{raw}} < 1.53 \times 10^{-3}$, $\text{FDR-P} < 1.53 \times 10^{-3}$) and increased RD (IVW β = 0.259, 95% CI = 0.111~0.406,

$P_{\text{raw}} < 5.84 \times 10^{-4}$, $\text{FDR-P} < 5.85 \times 10^{-4}$) of CGC, increased MD (IVW $\beta = 0.236$ 95% CI = 0.074~0.398, $P_{\text{raw}} < 4.22 \times 10^{-3}$, $\text{FDR-P} < 4.22 \times 10^{-3}$) and increased AD (IVW $\beta = 0.229$, 95% CI = 0.075~0.383, $P_{\text{raw}} < 3.60 \times 10^{-3}$, $\text{FDR-P} < 3.59 \times 10^{-3}$) in BCC.

Comment 1.2

-L158–159: Expand on the multivariate MR findings. Include statistical results including effect sizes and p-values for all associations tested. Additionally, in this context it is unclear what it means that "only PTSD was retained". Please clarify.

Response:

We sincerely thank the reviewer for their insightful comments and apologize for the lack of clarity in our original presentation. We acknowledge that our phrasing "only PTSD was retained" was ambiguous and failed to properly report key statistical metrics. While we intended to concisely present the core findings, we recognize this approach compromised conceptual precision and statistical transparency. We've comprehensively revised this section to:

- 1. Clarify the meaning of "retained" in the multivariate MR context:** The phrase now explicitly indicates that among multivariate MR analyses for SCZ, OUD, and PTSD, only PTSD showed statistically significant associations after rigorous testing.
- 2. Systematically report all relevant statistical indicators:** We have expanded the reporting to include: effect sizes with 95% CI, p-values for all associations, clearer context for BD validation outcomes

The revised text is as follows (the scarlet text represents represent the text added):

Text Revision 1: Added complete statistical reports to **Results**

Multivariate Mendelian randomization and validation

(Clean version: page 9-10, line 192-204; Marked version: page 11, line 216-229)

We conducted multivariate MR analysis²⁵ on SCZ, OUD, and PTSD, and only the association between FXST_RD and PTSD (OR = 1.063, 95% CI = 1.004~1.125, $P < 3.61 \times 10^{-2}$) was retained. More details can be found in **Table S23**. Subsequently, we partially validated our results using GWAS data for ADHD, AUD, BD, and SCZ from the FinnGen R11 database. The causal association between the MO of FX and ADHD in the forward MR analysis was replicated (OR = 1.806, 95% CI = 1.127~2.892, $P < 1.39 \times 10^{-2}$). Although BD did not yield identical replication results, it produced similar results: one s.d. increase in MO of FX (OR = 1.486, 95% CI = 1.065~2.072, $P < 1.97 \times 10^{-2}$) and RD of FXST (OR = 1.335, 95% CI = 1.025~1.740, $P < 3.21 \times 10^{-2}$) were associated with increasing risk of BD, and one s.d. increase FA of FXST (OR = 0.776, 95% CI = 0.613~0.982, $P < 3.50 \times 10^{-2}$) was associated with decreasing risk of BD. Taking into account the close spatial relationship (with a certain degree of overlap) between FX and FXST, as well as the potential correlations among the parameters FA, RD and MO, these similar results of BD could also be considered successful replication.

Comment 1.3 & 1.4

-L160 – 164: Similarly, please provide detailed replication results, including effect sizes, CIs, and p-values for each tested association and region. Clarify the meaning of “biologically comparable meanings” : what was tested to identify biological comparability? what were the results including metrics and p-values? Please also report which associations were found to be significant in the previous step but were deemed to not be “biologically meaningful” and why, providing the appropriate statistical results.

-L346 – 355: Please report all comparisons mentioned here explicitly and individually in the Results section, with their corresponding statistical results.

Response:

We sincerely appreciate the reviewer's critique regarding reporting completeness and terminology clarity. We acknowledge that our original phrasing “biologically comparable meanings” lacked sufficient definition, and we apologize for any confusion this introduced. This terminology was intended to describe a critical validation layer we applied beyond statistical significance thresholds to enhance biological plausibility.

Our validation approach operated on the principle that diffusion tensor imaging parameters (FA, MD, AD, RD, MO) exhibit neurobiologically constrained interrelationships within a given tract. Statistically significant results ($FDR-P < 0.05$) underwent secondary evaluation for directional concordance: specifically, whether parameter changes aligned with established neurobiological principles (e.g., FA decreases should correspond to MD/RD increases if reflecting genuine WMTs' deterioration). To operationalize this, we manually compared all nominally significant associations ($P_{raw} < 0.05$) for each implicated tract against our primary FDR-significant findings ($FDR-P < 0.05$), as documented in **Table S21**.

The phrase “biologically comparable meanings” referred specifically to concordant directional patterns across parameters within a tract (e.g., significantly decreased FA + increased MD/RD). Results showing statistically significant but biologically discordant parameter changes (e.g., significantly decreased FA + significantly decreased MD) were deemed less interpretable despite passing FDR correction. For transparency, we now explicitly call this process “**directional concordance validation**” throughout the manuscript.

In conclusion, we've implemented the following revisions to address your concerns:

1. **Added complete statistical reporting** (effect sizes, CIs, p-values) for all replication analyses.

2. Expanded and rewrote the *Method: Selection and Interpretation* section to make the statement of validation process more clear.

3. Reported all comparison outcomes individually in the Results section with corresponding statistics.

These modifications ensure full transparency about both statistically significant outcomes and their biological coherence assessments. We thank the reviewer for prompting this essential refinement to our analytical narrative.

The revised text is as follows (the scarlet text represents represent the text added):

Text Revision 1: Added complete statistical reports to *Results*

Multivariate Mendelian randomization and validation

(Clean version: page 9-10, line 192-204; Marked version: page 11, line 216-229)

We conducted multivariate MR analysis²⁵ on SCZ, OUD, and PTSD, and only the association between FXST_RD and PTSD (OR = 1.063, 95% CI = 1.004~1.125, $P < 3.61 \times 10^{-2}$) was retained. More details can be found in **Table S23**. Subsequently, we partially validated our results using GWAS data for ADHD, AUD, BD, and SCZ from the FinnGen R11 database. The causal association between the MO of FX and ADHD in the forward MR analysis was replicated (OR = 1.806, 95% CI = 1.127~2.892, $P < 1.39 \times 10^{-2}$). Although BD did not yield identical replication results, it produced similar results: one s.d. increase in MO of FX (OR = 1.486, 95% CI = 1.065~2.072, $P < 1.97 \times 10^{-2}$) and RD of FXST (OR = 1.335, 95% CI = 1.025~1.740, $P < 3.21 \times 10^{-2}$) were associated with increasing risk of BD, and one s.d. increase FA of FXST (OR = 0.776, 95% CI = 0.613~0.982, $P < 3.50 \times 10^{-2}$) was associated with decreasing risk of BD. Taking into account the close spatial relationship (with a certain degree of overlap) between FX and FXST, as well as the potential correlations among the parameters FA, RD and MO, these similar results of BD could also be considered successful replication.

Text Revision 2: Expanded and rewrote part of *Methods*

Selection and interpretation of results

(Clean version: page 17, line 415-424; Marked version: page 20, line 467-476)

Second, we established neurobiological coherence criteria for DTI parameters. Given their known physiological interrelationships, causal associations were required to demonstrate directional consistency across metrics/parameters within each WMT. Specifically: 1) Decreased FA must correspond with either increased MD and/or increased RD and/or decreased AD; 2) Increased MD must align with either increased RD or decreased AD. This approach prioritized findings with greater biological plausibility by evaluating directional concordance through established neurobiological mechanisms. For example, reduced FA accompanied by increased MD and RD would consistently indicate myelin impairment. While statistically significant but biologically discordant results (e.g., isolated parameter changes lacking supporting directional

patterns) would be dismissed, this tiered evaluation supplemented—rather than replaced—statistical significance thresholds.

Comment 1.5

-L357 – 361: *The sentence describing confounding factor assessment and manual descriptive interpretations is vague. Clarify whether these steps were part of the analysis pipeline or the discussion, and describe how empirical evidence was used for interpretation. Clearly state the number of tests performed per method for each trait pair. List all confounders identified and excluded, and explain how this influenced the results.*"

Response:

We sincerely appreciate the reviewer's guidance regarding methodological transparency in confounder assessment. We acknowledge our initial description lacked sufficient detail and have substantially revised both the methods and results sections to provide comprehensive clarification. Below are details of key revisions:

- 1. Expanded and rewrote the *Method: Selection and Interpretation* section: We provided a more detailed description of the assessment methods for confounding factors.** Briefly, confounding factor identification/assessment was an integral component of our analytical pipeline. We systematically retrieved all traits associated with each candidate SNP using LDlink, subsequently excluding SNPs unrelated to brain imaging or psychiatric disorders. This process identified 291 confounders across socioeconomic, lifestyle, and clinical domains (e.g., income, smoking, coronary artery disease, uterine fibroids), comprehensively documented in **Table S9**. To quantify impact, we:
 - Performed parallel MR analyses using both pre-exclusion (N=1,667 SNPs) and post-exclusion (N=1,376 SNPs) instruments for all pairs
 - Compared IVW estimates before/after exclusion in **Table S18**
 - Defined "significant confounder influence" as both:
 - a) Directional reversal of β -values
 - b) Change from non-significant to significant ($P > 0.05 \rightarrow P < 0.05$) or vice versa
- 2. Added a new section reporting the confounder analysis pipeline results.** In the **Results** → **Overview of MR** section, we have added detailed descriptions of the confounder selection process, providing readers with a complete list of all identified confounders (**Table S9**). Furthermore, the **"Confounding Factors, Outliers, and Sensitivity Analysis"** section originally in the **Supplementary Material** has been migrated to the **Results Section** and edited to enable more detailed reporting of results.
- 3. Terminology Revision:** We have eliminated the ambiguous "manual descriptive interpretations" designation. The entire confounder evaluation framework is now explicitly

documented in the **Methods ("Selection and interpretation of results")** and **Results ("Confounding factors, outliers, and sensitivity analysis")** sections, with full analytical outputs in **Tables S9, S10, S18**.

4. Additional response to "Empirical Evidence Integration": When confounder exclusion altered significance and direction of effect value, we examined trait-SNP relationships in existing literature to determine biological plausibility. However, in our analysis, no associations showed directional reversal. **We added this description in the Result Section.**

The revised text is as follows (the scarlet text represents represent the text added):

Text Revision 1: Expanded and rewrote "Selection and interpretation of results" in *Methods*

Selection and interpretation of results

The Original Text:

As Carter suggested, we should interpret and promote findings from MR studies with extreme caution⁵¹. Without rigorous validation and empirical verification, MR research is essentially a statistical exercise concerning genes, but we hoped that our results would be well-explained both statistically and biologically. Therefore, results meeting the following criteria were selected: (1) Results must be statistically significant and pass pleiotropy and heterogeneity tests, which means all results' $FDR_p < 0.05$, $MR_PPRESSO_Global_test\ p > 0.05$, $MR_egger_intercept\ p > 0.05$. Of the 4 methods used in the primary MR analysis, at least 3 must have results in the same direction; (2) Changes in different parameters of WMTs should be mechanistically plausible: this implies consistency in the changes of FA, AD, RD, and MD. Thus, when a WMT exhibited a causal association with a psychiatric disorder, we compared the data of its nominally significant FA, MD, AD, and RD parameters ($p < 0.05$) to obtain credibility for the neurobiological mechanism. Therefore, we ensured that the direction of change in FA is opposite to that of MD, the change in FA is opposite to at least one of the changes in RD and AD, and the change in MD is in the same direction as at least one of the changes in RD and AD. For instance, a decrease in FA (indicating loss of WMT integrity) theoretically corresponded to increases in MD (cellular damage) and RD (myelin sheath damage), and decrease in AD (axonal damage), which was explainable both neurobiologically and mathematically.

The Edited Text:

(Clean version: page 17, line 411-424; Marked version: page 19-20, line 463-476)

First, all reported results satisfied strict statistical criteria: 1) FDR-adjusted p-value < 0.05 ; 2) No evidence of pleiotropy ($MR_PRESSO\ Global\ Test\ p > 0.05$, $MR_Egger\ intercept\ p > 0.05$); and 3) Directional consistency across ≥ 3 of the 4 MR methods.

Second, we established neurobiological coherence criteria for DTI parameters. Given their known

physiological interrelationships, causal associations were required to demonstrate directional consistency across metrics/parameters within each WMT. Specifically: 1) Decreased FA must correspond with either increased MD and/or increased RD and/or decreased AD; 2) Increased MD must align with either increased RD or decreased AD. This approach prioritized findings with greater biological plausibility by evaluating directional concordance through established neurobiological mechanisms. For example, reduced FA accompanied by increased MD and RD would consistently indicate myelin impairment. While statistically significant but biologically discordant results (e.g., isolated parameter changes lacking supporting directional patterns) would be dismissed, this tiered evaluation supplemented—rather than replaced—statistical significance thresholds.

The Original Text:

Meanwhile, since we indiscriminately excluded SNPs related to non-brain imaging changes and psychiatric disorders when selecting IVs, to avoid overly stringent control due to our exclusions, we conducted MR separately with both pre- and post-exclusion IVs. We identified confounding factors that could reverse significance and performed manual descriptive interpretations to determine whether the results aligned with existing empirical evidence.

The Edited Text:

(Clean version: page 17, line 426-432; Marked version: page 20, line 478-484)

Third, we implemented a parallel-instrument analytical approach to address potential overcorrection bias from indiscriminate exclusion of SNPs unrelated to brain imaging or psychiatric disorders. First, parallel MR analyses were conducted using both pre-exclusion and post-exclusion IVs sets. When exclusion of confounder-associated SNPs materially altered results, which defined as both (1) reversal of β -value directionality and (2) transition across the statistical significance threshold ($P < 0.05$), we conducted literature search to exam whether there are other studies could support the alteration of results between the excluded traits and target outcomes.

Text Revision 2: Added detailed report of confounder assessment in *Results*

Overview of MR

(Clean version: page 4, line 94-99; Marked version: page 5-6, line 107-114)

...All Mendelian randomization (MR) processes followed the Burgess²³ and STROBE (Strengthening the Reporting of Observational Studies in Epidemiology) guidelines²⁴ and the checklist was in **Table S2**.

Before addressing confounding factors, we screened 1,667 SNP loci as instrumental variables for forward MR (**Table S5**) and 301 for reverse MR (**Table S6**), leaving 1,376 and 301 respectively after confounders removal (**Table S7 - S10**). Besides previously reported confounders such as socioeconomic status and

education, additional non-brain structural and non-psychiatric factors were included, such as pulse pressure (rs2645466), coronary artery disease (rs4894803), and uterine leiomyoma or ER-positive breast cancer (rs10828248), etc. (**Table S9**)...

Confounding factors, outliers, and sensitivity analysis

(Clean version: page 10, line 206-215; Marked version: page 12, line 231-240)

To quantify confounding effects, we conducted parallel MR analyses using both pre-exclusion (N=1,667 SNPs) and post-exclusion (N=1,376 SNPs) IVs across all pairs. The comparative results (**Table S9 and S18**) revealed that only 2 associations exhibited meaningful changes of IVW estimates before/after confounder adjustment (among the 24 statistically significant associations): FX_MO - ADHD and GCC_MD - MDD transitioned from non-significant to significant ($P > 0.05 \rightarrow P < 0.05$), while all others maintained consistent directionality and significance thresholds. Specifically, 1 SNP related to ascending thoracic aortic diameter was removed from FX_MO. And 6 SNPs, primarily associated with lipids, Alzheimer's disease, aging, and glutamic-oxaloacetic transaminase levels, were removed from the IVs of GCC_MD. Crucially, no associations showed β -value reversals.

Comment 2: Sample Overlap and Bias Assessment

"The manuscript states that datasets with less than 5% sample overlap were included, estimating a potential bias of <0.15%. However, this point needs further elaboration. The authors should:

-Discuss in which ways this overlap might still bias MR estimates (eg direction of bias).

-Include a sensitivity analysis using MRlap, an MR method that explicitly accounts for sample overlap, particularly for AUD and PTSD."

Response:

We sincerely appreciate the reviewer's insightful suggestions regarding sample overlap considerations. We have now implemented comprehensive MRlap analyses (Mounier & Kutalik, 2023) to quantitatively evaluate potential bias, with full methodological details added to our sensitivity analysis section and results documented in **Tables S22**.

1. Key Findings from MRlap Analysis: Among the 24 significant WMTs-psychiatric disorder associations identified in our primary analysis:

- 16 demonstrated detectable sample overlap influence (P difference < 0.05)
- Crucially, all 24 associations maintained consistent effect directionality after bias adjustment
- Notably, corrected effect estimates consistently strengthened rather than attenuated toward null (absolute β -values increased)

2. Technical Note: As MRlap requires raw summary statistics without confounder/outlier filtering, the IV sets differ from our primary analysis. While this precludes direct effect-size comparisons, the complete directional concordance between MRlap-adjusted and primary results provides robust validation against sample overlap bias. We have transparently **documented this limitation in our *Discussion* section.**

3. Results of MRlap:

Exposure	Outcome	Observed_effect	Corrected_effect	p_difference
FX_MO	ADHD	0.05	0.07	5.57E-02
IFO_AD	AUD	(0.09)	(0.13)	7.76E-08
FXST_MO	BD	(0.08)	(0.11)	1.03E-05
GCC_MD	MDD	0.02	0.02	5.49E-02
GCC_MD	ODD	(0.08)	(0.10)	8.23E-03
GCC_RD	ODD	(0.06)	(0.09)	6.21E-02
ACR_MO	PTSD	(0.02)	(0.03)	3.82E-01
CGH_MD	PTSD	0.02	0.03	1.83E-02
CGH_RD	PTSD	0.02	0.03	3.06E-02

FXST_RD	PTSD	0.04	0.06	1.12E-02
PCR_AD	PTSD	0.04	0.06	1.29E-05
PCR_FA	PTSD	(0.02)	(0.03)	1.36E-03
PCR_RD	PTSD	0.03	0.04	1.18E-02
RLIC_RD	PTSD	0.04	0.05	6.48E-05
SFO_FA	PTSD	(0.04)	(0.05)	5.56E-04
SFO_MD	PTSD	0.02	0.03	6.15E-02
SLF_FA	PTSD	(0.02)	(0.03)	3.27E-02
SLF_RD	PTSD	0.02	0.02	3.81E-02
UNC_FA	PTSD	(0.02)	(0.03)	1.75E-01
UNC_MD	PTSD	0.02	0.03	1.07E-01
FXST_FA	SCZ	(0.05)	(0.07)	3.54E-01
RLIC_MO	SCZ	0.07	0.10	2.25E-04
SCR_FA	SCZ	0.05	0.07	2.94E-02
FXST_MO	TS	(0.51)	(0.75)	5.70E-08

The revised text is as follows (**the scarlet text represents represent the text added**):

Text Revision 1: Added explicit description of MRlap and interpretation framework to *Methods*

Sensitivity analysis and outlier screening

(Clean version: page 16, line 401-402; Marked version: page 19, line 437)

We employed MR-PRESSO, MR Egger intercept, Cochran's Q statistic^{48,49}, and leave-one-out analysis for sensitivity and pleiotropy assessments. **And we used MRlap to assess the influence of sample overlap**⁵⁰.

Text Revision 2: Incorporated key findings with directional concordance emphasis to *Results*

Confounding factors, outliers, and sensitivity analysis

(Clean version: page 10, line 227-232; Marked version: page 12-13, line 252-257)

... Furthermore, among the 24 significant WMTs-psychiatric disorder associations, 16 showed evidence of sample overlap influence ($p_{\text{difference}} < 0.05$). Crucially, all associations maintained consistent β -value directionality before and after sample overlap adjustment (**Table S22**). Notably, corrected effect estimates did not attenuate toward the null point; instead, effects strengthened through increased absolute β -values. This directional reinforcement aligns with and further substantiates our primary findings. Finally, we observed coordinated alterations across all diffusion tensor imaging parameters. These findings collectively confirm the reliability of our MR causal inferences (please see the details in **Table S20**).

Text Revision 3: Added nuanced interpretation of MRlap results acknowledging technical

constraints while highlighting confirmatory value to *Discussion*

(Clean version: page 13-14, line 310-315; Marked version: page 16, line 345-350)

This study had several limitations. Firstly, some overlap (<5%) was unavoidable in the samples, potentially introducing bias into our results. Although we used MRlap analysis to evaluate the effect of sample overlapping, MRlap could only evaluate the aggregate effects of sample overlap on results (indicating bias direction) without enabling precise quantification of bias effect size. Given that the MRlap findings ultimately aligned with our primary results, we conclude that while sample overlap introduced detectable bias, such bias would not alter the direction or attenuate the magnitude of our effect estimates ...

Supplementary Table Revision: The results of MRlap have been added in Table S22.

New reference added:

Mounier, N. & Kutalik, Z. Bias correction for inverse variance weighting Mendelian randomization. *Genet Epidemiol* **47**, 314-331 (2023).

Comment 3: Mini Review Section

"The purpose and placement of this section are unclear. It is not introduced as part of the introduction, and although search terms used are provided, there is no description of screening criteria, quality assessment, or study selection, making the reporting and methods used insufficient for a review. However, the authors did search the literature extensively, and it is worth reporting their findings outside the context of a "mini review. Thus, we recommend:

-Please remove Mini Review from the Results and integrate relevant insights into the Discussion. If deemed necessary, methodological details for the literature search conducted may be moved to the Supplementary Materials."

Response:

We sincerely appreciate the reviewer's discerning assessment regarding the placement and methodology of the mini-review section. We agree that this content was better suited to contextualize findings in the **Discussion** rather than presenting it as a standalone results component. Our original intent was to provide readers with comprehensive background on established WMTs-psychiatric disorder relationships before introducing our novel causal findings. However, we recognize that its positioning in the **Results** section without full systematic review methodology was suboptimal. Thus, following reviewer's insights, we implement revisions:

1. The entire **Mini-Review** section has been **removed** from Results.
2. Core insights on WMT-pathology relationships have been integrated into the Discussion to contextualize our novel findings.
3. The **methodology** of **Mini-Review** has been **migrated** to **Supplemental Materials**.
4. Relevant citations supporting established WMT-disorder relationships are now strategically positioned as supporting evidence in the Discussion.

The revised text is as follows (**the scarlet text represents represent the text added**):

Text Revision 1: Added part information of "Mini-Review" to **Discussion**

(Clean version: page 13, line 292-294; Marked version: page 15, line 327-329)

... Thus, alterations in FXST may contribute to emotional dysregulation^{28,29} (in SCZ, BD, TS, related to the amygdala) or cognitive impairments^{28,29} (in ADHD, related to the hippocampus). We speculated that the similar results of ADHD, TS, BD and SCZ are partly coming from the overlap of their similar pathogenesis, **as suggested by evidence linking WMTs integrity in these regions to both genetic influences and environmental exposures, particularly in psychiatric developmental disorders like ADHD and ASD^{4,30}, which** could partly explain the similarity of some clinical symptom (such as mood instability).

Comment 4: Appropriate citation of prior research

" In L74, the authors claim that no association has been established between white matter tracts and psychiatric disorders in MR. However, Brainwide MR for anxiety disorders has yielded a positive association of white matter hyperintensities with anxiety disorders. Please refer to this in the introduction."

Response:

We sincerely thank the reviewer for identifying this important gap in our literature coverage. We appreciate the opportunity to acknowledge the valuable work by Zanoaga et al. (2024) that establishes causal links between white matter hyperintensities and anxiety disorders using MR methodology. We have integrated this seminal study within the broader context of neuroimaging-MR literature. The added reference (Zanoaga et al., *Biol Psychiatry* 2024) appropriately contextualizes our work while highlighting the novelty of our WMTs-specific diffusion metric approach. We are grateful for this constructive suggestion which strengthens our scholarly framing.

The revised text is as follows **(the scarlet text represents represent the text added)**:

Text Revision 1: Added new reference in to **Introduction**

(Clean version: page 4, line 73-74; Marked version: page 5, line 86-87)

Current MR research has explored the causal association between imaging-derived phenotype and 10 psychiatric disorders¹⁸, between brain functional networks and 12 psychiatric disorders¹⁹, **between white matter hyperintensities and anxiety disorders²⁰**, between brain structure and Alzheimer's disease (ALZ)²¹, and between cortical structure, white matter microstructure, and neurodegenerative diseases²².

New reference added:

Zanoaga, M.D., *et al.* Brainwide Mendelian Randomization Study of Anxiety Disorders and Symptoms. *Biol Psychiatry* **95**, 810-817 (2024).

Minor Phrasing Issues

-L42: Rephrase to: "...which shows that eight psychiatric disorders are among the top 25 contributors to global disease burden."

-L74: Omit "either" from this sentence to help with clarity.

-L119: Remove the word "between".

-L198–199: Replace "anxious" with "generalized anxiety disorder" for clarity and precision.

Response:

We sincerely appreciate the reviewer's meticulous attention to linguistic precision, which has significantly enhanced the clarity and professionalism of our manuscript. The suggested phrasing improvements have been fully implemented in our manuscript, please check it.

The revised text is as follows **(the scarlet text represents represent the text added)**:

Text Revision 1: Rephrase the sentence in **Introduction**

The Original Text:

Psychiatric disorders impose a substantial burden on global health, as evidenced by the Global Burden of Diseases (GBD) 2021¹, which highlights 8 psychiatric disorders among the top 25 disease burdens.

The Edited Text:

(Clean version: page 3, line 43-45; Marked version: page 4, line 55-57)

Psychiatric disorders impose a substantial burden on global health, as evidenced by the Global Burden of Diseases (GBD) 2021¹, which shows that 8 psychiatric disorders are among the top 25 contributors to global disease burden.

Text Revision 2: Omit "either" in to help with clarity

(Clean version: page 4, line 77; Marked version: page 5, line 90)

However, these studies ~~either~~ fail to uncover the causalities between WMTs and psychiatric disorders.

Text Revision 3: Remove the word "between"

(Clean version: page 6, line 143; Marked version: page 8, line 162)

... as well as of the inferior fronto-occipital fasciculus (IFO) with ~~between~~ AUD.

Text Revision 4: Replace "anxious" with "generalized anxiety disorder" for clarity and precision

(Clean version: page 12, line 258; Marked version: page 14, line 291)

For example, ~~general-anxious-disorder~~ ~~generalized anxiety disorder~~ and panic disorders are classified equally as anxious disorders in the database.

The Reply to Reviewers

Reviewer #2

General Response to Reviewer #2

Dear Reviewer #2,

We are deeply grateful for the insightful recommendations provided by you and we extend our sincere appreciation for your contributions.